# Beyond Freezing the Router: Rank-Aligned Post-Training Quantization for Mixture-of-Experts Models

**Yi-Zeng Fang**                                                   *joycefang1213.ee11@nycu.edu.tw*
*Institute of Electronics, National Yang Ming Chiao Tung University*

**Juinn-Dar Huang**                                                *jdhuang@nycu.edu.tw*
*Institute of Electronics, National Yang Ming Chiao Tung University*

**Reviewed on OpenReview:** *https://openreview.net/forum?id=bPsPPI65hf*

## Abstract

Quantizing Mixture-of-Experts language models remains a challenging problem because quantization noise propagates across layers and distorts downstream expert selection. Although common practice keeps the router in full precision, we show that this strategy is insufficient: quantization-induced errors in expert outputs still shift the logits of the next-layer router, and freezing the router removes the opportunity to compensate for these shifts. Motivated by this finding, we propose **RouteQuant**, a post-training quantization framework that explicitly embraces *router quantization* to correct for expert-level distortion. We analyze how quantization alters router rankings and rank flips, and provide a theoretical proof showing that deviations in expert outputs are bounded by both expert-selection and gap-preservation errors. These insights motivate two router-alignment objectives: (i) *Rank-Aware Jaccard Loss*, which aligns the top-$k$ routing sets between full-precision and quantized models, and (ii) *Gap Hinge Loss*, which preserves the margin between consecutive expert logits to suppress rank flipping. In addition to router alignment, we further introduce *Expert-Aware Smoothing Factor*, which assigns separate activation smoothing factors to heterogeneous experts. Across OLMoE, DeepSeek-MoE, and Qwen3-MoE, RouteQuant consistently improves perplexity on C4 and WikiText-2 and enhances zero-shot accuracy under W4A4 and W4A8 across diverse downstream tasks, demonstrating the effectiveness of the proposed framework.

## 1 Introduction

Large language models (LLMs) continue to advance rapidly and reshape modern natural language processing (Achiam et al., 2023; Grattafiori et al., 2024; Guo et al., 2025; Yang et al., 2025). As parameter counts and training corpora grow, Mixture-of-Experts (MoE) architectures emerge as a scalable design that raises effective capacity without proportional compute (Shazeer et al., 2017; Fedus et al., 2022; Dai et al., 2024; Muennighoff et al., 2025). An MoE layer comprises a learned router and a pool of experts; for each input token, the router computes routing scores, activates the top-$k$ experts, and aggregates their outputs. Variants include shared experts that capture common knowledge across tokens, while routed experts specialize. As model size and MoE adoption increase, deployment becomes constrained by memory and latency, so low-precision inference via quantization becomes essential for practical serving.

Quantizing MoE models introduces challenges beyond those observed in dense Transformers, as each layer's output is jointly shaped by the router's expert selection and the experts' transformed activations. Prior studies have highlighted the router's sensitivity, noting that even minor perturbations can lead to misrouting (Ma et al., 2025; Fu et al., 2025). Since the router accounts for only a tiny fraction of total model parameters (*e.g.*, 0.03%), some PTQ systems keep router layers in full precision to avoid degradation and preserve routing accuracy (Kwon et al., 2023; Chen et al., 2025a). However, keeping the router weights in full

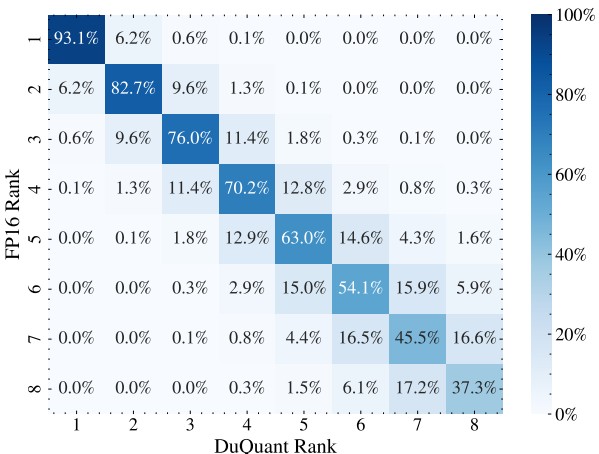

Figure 1: Confusion matrices at **layer 0** comparing FP16 vs. DuQuant (Lin et al., 2024a) top-$k$ indices under W4A4 without quantizing router.

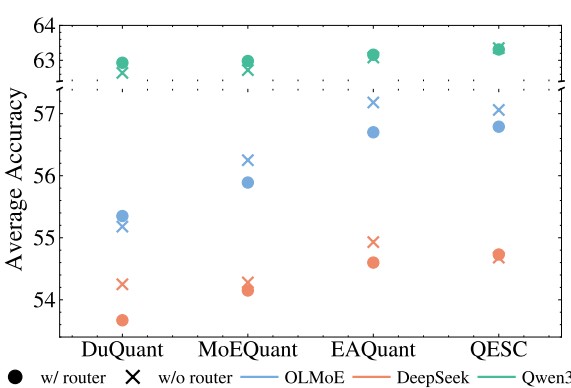

Figure 2: Effect of router quantization under W4A4. **w/ router** (•) quantizes the router together with experts, while **w/o router** (×) keeps the router in full precision.

precision does not necessarily preserve the original FP16 routing behavior after quantization. Quantization of preceding layers and MoE blocks can perturb the hidden states passed to later routers; consequently, even a full-precision router may receive inputs that differ from those in the FP16 model and produce different top-$k$ expert selections. We illustrate this routing mismatch in Figure 1. Importantly, our goal is not to learn an unconstrained new routing policy for the quantized experts. Instead, we aim to preserve the FP16 routing behavior as much as possible under quantization-induced activation perturbations. As a first observation, we find that simply quantizing the router can sometimes improve MoE PTQ performance compared with keeping it in full precision, as shown in Figure 2. This suggests that the router need not always remain full precision after expert quantization. Yet this observation also raises a key question: if plain router quantization is only sometimes helpful, what aspects of routing should be preserved or controlled to make router quantization reliable?

To answer this question, we analyze how routing perturbations affect MoE outputs and identify two controllable factors: (i) the selected expert indices, which determine which experts contribute to the output, and (ii) the inter-expert logit gaps, which determine how stable the top-$k$ routing decision is under perturbations. This analysis shows that effective router quantization should not merely reduce router precision, but should preserve the FP16 top-$k$ structure and the margins that protect it from rank flipping. Guided by this insight, we introduce two router-alignment objectives. *Rank-Aware Jaccard Loss (RAJ)* aligns the top-$k$ expert identities and their rank-aware structure, while *Gap Hinge Loss (GH)* preserves the score margins between adjacent experts to reduce rank flipping near the top-$k$ boundary. These objectives use the FP16 router behavior as the reference and encourage the quantized router to recover stable FP16 routing under quantized activations. In addition, we incorporate an *Expert-Aware Smoothing Factor (ES)* that assigns each expert a channel-wise smoothing factor, preventing heavy-tailed experts from distorting shared quantization ranges and further reducing expert-side quantization noise. Together, RAJ, GH, and ES form a cohesive MoE-aware PTQ strategy that jointly addresses routing mismatch and expert-output distortion.

Our main contributions are as follows:

- We show that keeping router weights in full precision does not necessarily preserve FP16 routing under quantized computation, since quantization perturbs the hidden states passed to later routers. We therefore propose router-aware calibration to recover FP16 routing behavior under quantized activations.

- We theoretically link router quantization error to expert output error and identify two key factors, the selected expert indices and the inter-expert score gap. Accordingly, we introduce *Rank-Aware Jaccard Loss* and *Gap Hinge Loss* to stabilize both.

- We validate our framework on OLMoE, DeepSeek-MoE, and Qwen3-MoE, achieving lower perplexity on C4 and WikiText-2 and higher accuracy on diverse reasoning tasks.

## 2 Related Work

### 2.1 Mixture-of-Experts LLMs

Early studies propose using gating networks to adaptively route each input to specialized sub-networks (Jacobs et al., 1991; Jordan & Jacobs, 1994), and subsequent work extends this idea to a variety of domains (Deisenroth & Ng, 2015; Aljundi et al., 2017). In LLMs, an MoE layer places expert MLPs behind a lightweight gate (linear projection plus softmax) and routes each token to the top-$k$ experts with load-balancing regularization (Shazeer et al., 2017); systems advance scale MoE transformers with automated sharding and parallelism (Lepikhin et al., 2021).

Large-scale instances vary in routing. Switch transformer uses top-1 gating to reduce activation cost (Fedus et al., 2022); GLaM shows that top-2 improves the accuracy, an efficiency trade-off at trillion parameter scale (Du et al., 2022); Mixtral 8×7B activates two experts per token and rivals dense peers at similar cost (Jiang et al., 2024). DeepSeek-MoE increases expert granularity, keeps a few active experts per token, and adds always-on shared experts to capture global knowledge (Dai et al., 2024); DeepSeek-V2/V3 further refine routing, optimization, and systems (Liu et al., 2024a;b).

Unlike dense transformers, which apply all parameters to every token, MoE models rely on a router to decide which subset of experts is activated. This router is therefore central to both efficiency and accuracy: small perturbations in its outputs directly change expert selection and propagate through the entire forward pass. Our work focuses on this router behavior and its interaction with expert heterogeneity, distinguishing MoE-specific challenges from those in dense architectures.

### 2.2 Post-Training Quantization for LLMs

PTQ is a standard route to deploy LLMs efficiently, reducing memory and bandwidth without retraining. In dense transformers, PTQ minimizes layerwise reconstruction error while preserving numerical structure relevant to generation. GPTQ casts weight-only quantization as a blockwise least-squares problem with Hessian-aware error compensation, delivering strong 4-bit accuracy at negligible calibration cost (Frantar et al., 2022). AWQ accounts for activation statistics during calibration, preserves high-saliency channels, and uses data-aware scaling to control outliers (Lin et al., 2024b). DuQuant redistributes activation outliers via a dual transformation that rebalances ranges in both activation and weight, enabling competitive W4A4 across dense LLMs (Lin et al., 2024a).

In MoE architectures, the router is a critical component because its outputs determine which experts are activated; even small perturbations in router logits can change the top-$k$ expert set and affect downstream accuracy. Existing MoE-specific PTQ methods address this challenge from different perspectives: MoEQuant reweights expert calibration by router weights but does not explicitly correct quantization-induced routing changes (Chen et al., 2025b); EAQuant quantizes the router and aligns router logit distributions with a KL-divergence objective (Fu et al., 2025); and QESC aligns the logits of highly probable experts with a top-$k$ MSE loss (Chen et al., 2025a). Thus, prior methods may either keep routers in full precision or include router quantization, depending on the system design. Rather than framing router quantization as simply replacing a full-precision router, RouteQuant improves router-quantized MoE PTQ by explicitly modeling two routing factors that are crucial under quantization: top-$k$ expert identity/order and inter-expert logit gaps, motivating router-aware calibration that preserves FP16 routing behavior under quantized router.

## 3 Methodology

### 3.1 Router–Expert Coupling in MoE Outputs

An MoE layer with $E$ experts and top-$k$ routing computes, for an input $x$,

$$y(x) \;=\; \sum_{j \in \text{top-}k} \pi_j(x)\,\mathcal{E}_j(x), \tag{1}$$

where $\pi_j(x)$ are routing weights derived from router logits $r(x) \in \mathbb{R}^E$, and $\mathcal{E}_j$ denotes the $j$-th expert. We consider a full-precision router $r^{(\text{fp})}$ and its quantized counterpart $r^{(q)}$. Let:

$$I = (i_1, \ldots, i_k) = \text{top-}k\big(r^{(\text{fp})}(x)\big), \tag{2}$$

$$J = (j_1, \ldots, j_k) = \text{top-}k\big(r^{(q)}(x)\big), \tag{3}$$

be the ordered top-$k$ expert indices for the full-precision and quantized routers, respectively. The corresponding MoE outputs are:

$$y^{(\text{fp})}(x) = \sum_{j \in I} \pi_j^{(\text{fp})}(x)\,\mathcal{E}_j(x), \tag{4}$$

$$y^{(q)}(x) = \sum_{j \in J} \pi_j^{(q)}(x)\,\mathcal{E}_j(x). \tag{5}$$

The two ordered top-$k$ lists $I$ and $J$ can differ in a simple but important way. Some experts are selected by both routers, some experts selected by the full-precision router disappear after quantization, and some new experts enter the quantized top-$k$ set. We therefore decompose the selected expert sets as:

$$S^* = I \cap J, \quad S^- = I \setminus J, \quad S^+ = J \setminus I. \tag{6}$$

Here, $S^*$ contains the shared experts selected by both routers, $S^-$ contains the full-precision selected experts dropped by the quantized router, and $S^+$ contains the newly selected experts introduced by the quantized router. This decomposition separates routing perturbations into two interpretable parts: changes on the shared experts and substitutions between dropped and newly selected experts, which we analyze next.

**Proposition 1** (Router-induced MoE output error)**.** *Assume that for a given input $x$, the expert outputs are uniformly bounded, i.e., $\|\mathcal{E}_j(x)\| \leq B(x)$, $\forall j$. Then the router-induced output discrepancy obeys:*

$$\big\|y^{(\text{fp})}(x) - y^{(q)}(x)\big\| \leq B(x)\Big( \sum_{j \in S^*} \big|\pi_j^{(\text{fp})}(x) - \pi_j^{(q)}(x)\big| + \sum_{j \in S^-} \pi_j^{(\text{fp})}(x) + \sum_{j \in S^+} \pi_j^{(q)}(x) \Big). \tag{7}$$

The first term in Equation (7) measures how much router weights on *shared* experts change; the latter two terms depend on the probability mass assigned to *mismatched* experts, *i.e.*, how much the top-$k$ sets differ. Thus, router-induced error is governed by two key factors: (i) which experts appear in the top-$k$ list, and (ii) how router weights change on the corresponding experts.

Next, we connect index changes to the underlying router logits. Let:

$$r_{(1)}^{(\text{fp})}(x) \geq \cdots \geq r_{(E)}^{(\text{fp})}(x), \tag{8}$$

denote the sorted full-precision logits, and define the consecutive logit gaps:

$$\Delta_r^{(\text{fp})}(x) = r_{(r)}^{(\text{fp})}(x) - r_{(r+1)}^{(\text{fp})}(x). \tag{9}$$

**Proposition 2** (Gap condition for top-$k$ stability)**.** *If the quantized consecutive gaps satisfy:*

$$\Delta_r^{(q)} \geq \Delta_r^{(\text{fp})}, \tag{10}$$

*for all $r \in \{1, \ldots, k\}$, then the ordered top-$k$ expert indices are preserved.*

Proposition 2 provides a sufficient condition for preserving the ordered top-$k$ routing under quantization. Specifically, if the quantized inter-expert logit gaps are no smaller than their full-precision counterparts, rank inversions cannot occur, and the selected experts remain unchanged. Combining Proposition 1 and 2, the router-induced MoE error is dominated by:

1. the *top-k index mismatch* between $I$ and $J$, and

2. the *logit gaps* $\Delta_r^{(\text{fp})}(x)$ that determine how easily the ranks flip.

The full proof is provided in Appendix A. This analysis motivates a router-alignment objective that explicitly stabilizes both the selected expert identities and the relative score gaps among experts. Based on this insight, we introduce two complementary loss functions that directly target these two sources of routing error. Notably, our quantization is built upon DuQuant (Lin et al., 2024a), and we integrate the proposed losses into the learnable weight clipping (Shao et al., 2024).

### 3.2 Rank-Aware Jaccard Loss (RAJ)

Proposition 1 shows that index mismatch between $I$ and $J$ directly inflates the router-induced error via the $S^-$ and $S^+$ terms in Equation (7). Empirically, we also observe that quantization errors in the router are not arbitrary: the quantized router usually selects experts that are close in rank to the full-precision choices, with most mistakes occurring near the top-$k$ boundary (see Figure 1). This suggests that aligning the *set and order* of selected experts, especially at higher ranks, is more important than matching all experts.

To capture this, we design a rank-weighted similarity objective on the top-$k$ expert indices. We first assign geometric weights $w_r = \beta^{r-1}$ with $\beta \in (0,1]$ so that higher ranks receive larger emphasis. We emphasize higher-ranked experts, as, from the router's perspective, they receive larger routing weights and therefore contribute more strongly to the final MoE output. Errors at these positions have a large impact: if a top-ranked expert is misrouted, the resulting representation can deviate substantially from the full-precision behavior. Using these, we define affinity vectors $A_{\text{fp}}, A_q \in \mathbb{R}^E$:

$$A_{\text{fp(q)}}(e) = \begin{cases} w_r, & \text{if } e = i_r(j_r), \\ 0, & \text{otherwise}, \end{cases} \tag{11}$$

The *Rank-Aware Jaccard (RAJ)* similarity between $I$ and $J$ is:

$$J_W(I, J) = \frac{\sum_{e=1}^E \min\big(A_{\text{fp}}(e), A_q(e)\big)}{\sum_{e=1}^E \max\big(A_{\text{fp}}(e), A_q(e)\big)} \in [0, 1],$$

and the corresponding loss is:

$$\mathcal{L}_{\text{RAJ}} = 1 - J_W(I, J). \tag{12}$$

This design directly measures the agreement between full-precision and quantized top-$k$ experts while prioritizing higher ranks. Unlike logit-based losses, it is invariant to affine transformations of $r$ that preserve ordering, and it decreases whenever near-miss errors (such as swapping rank-$k$ and rank-$(k+1)$ experts) are corrected. By maximizing overlap between $I$ and $J$, $\mathcal{L}_{\text{RAJ}}$ explicitly reduces the $S^-$ and $S^+$ contributions in Equation (7), thereby mitigating index-driven routing errors.

### 3.3 Gap Hinge Loss (GH)

While *RAJ* stabilizes which experts appear in the top-$k$ list, it does not control how close router scores are between adjacent experts, and thus cannot prevent rank flipping, as discussed in Proposition 2. To address this, we explicitly encourage larger gaps between consecutive experts within the full-precision top-$k$ sequence.

We first extract paired full-precision and quantized scores on the fixed full-precision top-$k$ index set $I = (i_1, \ldots, i_k)$:

$$r_r^{(\text{fp})} = r_{i_r}^{(\text{fp})}, \qquad \tilde{r}_r^{(q)} = r_{i_r}^{(q)}. \tag{13}$$

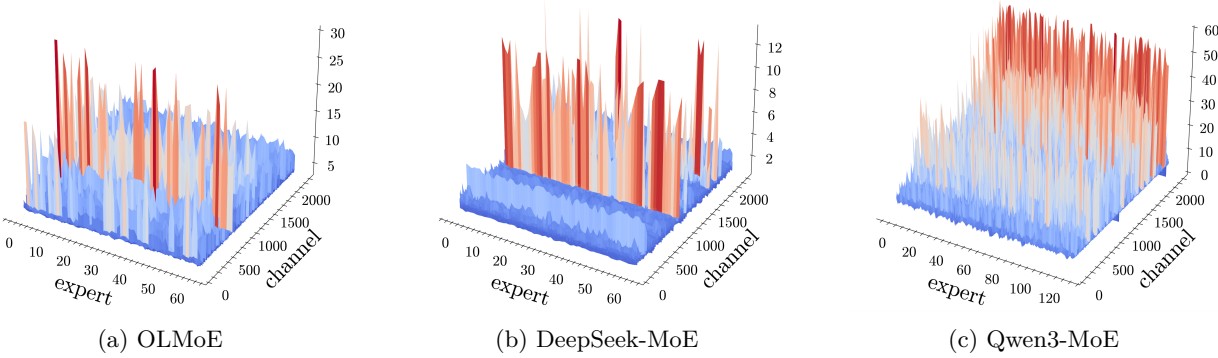

| (a) OLMoE | (b) DeepSeek-MoE | (c) Qwen3-MoE |

Figure 3: Expert activations in the **last layer** for each model.

We then define consecutive gaps as:

$$\Delta_r^{(\text{fp})} = r_r^{(\text{fp})} - r_{r+1}^{(\text{fp})}, \qquad \Delta_r^{(q)} = \tilde{r}_r^{(q)} - \tilde{r}_{r+1}^{(q)},$$

for $r = 1, \ldots, k-1$. The GH loss penalizes quantized gaps that shrink below the full-precision reference (up to a margin $\gamma \geq 0$):

$$\mathcal{L}_{\text{GH}} = \frac{1}{k-1} \sum_{r=1}^{k-1} \left[ \Delta_r^{(\text{fp})} - \Delta_r^{(q)} + \gamma \right]_+, \tag{14}$$

where $[x]_+ = \max(0, x)$. Since our target is to preserve the margins, we simply set $\gamma = 0$.

By preserving the logit gaps, $\mathcal{L}_{\text{GH}}$ enforces a condition of Equation (10): when $\Delta_r^{(q)}$ does not shrink too much relative to $\Delta_r^{(\text{fp})}$, rank inversions on the top-$k$ set become less likely under bounded quantization noise. Hence, *GH* reduces the probability of routing changes that would increase the index mismatch terms in Equation (7).

**Total router-consistency objective.** We combine *RAJ* and *GH* into a router-consistency objective:

$$\mathcal{L}_{\text{router}} = \lambda_{\text{RAJ}} \mathcal{L}_{\text{RAJ}} + \lambda_{\text{GH}} \mathcal{L}_{\text{GH}}, \tag{15}$$

Together, they implement the theoretical insights of Proposition 1 and 2 by jointly controlling top-$k$ indices and inter-expert margins.

### 3.4 Expert-Aware Smoothing Factor (ES)

The router-consistency objective targets expert *selection*, but MoE output error also comes from quantization noise in the experts themselves. To reduce this expert-side error, we introduce an *Expert-Aware Smoothing Factor (ES)* that assigns each expert its own channel-wise scaling. Existing multi-expert schemes typically compute a per-channel smoothing factor for each expert and then take the maximum across experts, forcing all experts in the layer to share one smoothing factor per channel (Fu et al., 2025). This max-aggregation is dominated by a few heavy-tailed experts and either clips light-tailed experts or wastes quantization levels on most experts. As seen in Figure 3, experts exhibit markedly different activation and weight statistics, motivating an expert-specific design. Additional layers are shown in Figures 11 to 13.

Let an MoE MLP take input $\boldsymbol{x} \in \mathbb{R}^d$, and let expert $i \in \{1, \ldots, E\}$ use weights $W^i \in \mathbb{R}^{m \times d}$ with column $\mathbf{W}_j^i$ for channel $j$. Following SmoothQuant (Xiao et al., 2023), we introduce a positive diagonal matrix $D_i = \text{diag}(\boldsymbol{s}^i)$, where $\boldsymbol{s}^i = (s_1^i, \ldots, s_d^i)$, and rewrite:

$$W^i \boldsymbol{x} = \left( W^i D_i \right) \left( D_i^{-1} \boldsymbol{x} \right) \triangleq \widetilde{W}^i \widetilde{\boldsymbol{x}}^i, \tag{16}$$

which is algebraically exact in full precision. We then quantize $\widetilde{W}^i$ per channel and $\widetilde{\boldsymbol{x}}^i$ per token.

Table 1: Reported total parameters, active parameters, and expert configuration for all models.

| Model | Total Param. | Act. Param. | top-$k$ E. | Total E. |
|---|---|---|---|---|
| OLMoE | 6.7B | 1.3B | 8 | 64 |
| DeepSeek | 16.4B | 2.8B | 6 | 64 |
| Qwen3 | 30.5B | 3.3B | 8 | 128 |

Let $|x_j|$ denote the magnitude of the $j$-th input activation on a small calibration set, and $|\mathbf{W}_j^i|$ the absolute values of the corresponding weight column. For $j = 1, \ldots, d$, $i = 1, \ldots, E$, and $\alpha \in [0, 1]$, we define:

$$s_j^i = \frac{\max(|x_j|)^\alpha}{\max(|\mathbf{W}_j^i|)^{1-\alpha}}, \tag{17}$$

for normalization between activations and weights.

Compared with a single shared $s_j$ per channel, *ES* adapts to each expert's statistics, reduces activation clipping, and stabilizes weight quantization. This improves expert outputs that the router aggregates, further lowering MoE output logit error with negligible runtime overhead as shown in Table 7.

## 4 Experiments

### 4.1 Settings

**Setup.** We conduct all experiments on a single NVIDIA A100 (80 GB) GPU using `PyTorch` and fix the random seed across runs. For post-training calibration, we sample 256 sequences from the C4 corpus with a sequence length of 2048 tokens (Raffel et al., 2020). We use DuQuant (Lin et al., 2024a) as our underlying quantization framework and apply per-token activation quantization and per-channel weight quantization, following Fu et al. (2025). We configure the router to W8A8 in all experiments and ablation studies unless otherwise specified. The hyperparameters are set to $\alpha = 0.6$, $\beta = 0.95$, $\lambda_{\text{RAJ}} = 1$ and $\lambda_{\text{GH}} = 1$. We evaluate our method on state-of-the-art MoE models: OLMoE (Muennighoff et al., 2025), DeepSeek-MoE (Dai et al., 2024), and Qwen3-MoE (Yang et al., 2025). The detailed model configurations are summarized in Table 1.

**Dataset.** We evaluate perplexity on WikiText-2 (Merity et al., 2017) and C4 (Raffel et al., 2020). For zero-shot accuracy, we use ARC-Challenge, ARC-Easy (Clark et al., 2018), BoolQ (Clark et al., 2019), HellaSwag (Zellers et al., 2019), OpenBookQA (Mihaylov et al., 2018), RTE (Dagan et al., 2005), and WinoGrande (Sakaguchi et al., 2021). We follow the standard zero-shot protocol (no in-context examples) and score multiple-choice options by their average token log-likelihood. In addition, we evaluate our method on math and coding benchmarks, with details provided in Appendix B.

**Evaluation Metrics.** We evaluate models using token-level perplexity, zero-shot accuracy, and router consistency. Perplexity measures the model's average surprise over the evaluation corpus, where lower values indicate better language modeling; accuracy measures the fraction of correct predictions on downstream tasks, where higher values are better. We compute perplexity with model's native tokenizer and measure accuracy under the standard zero-shot protocol. For zero-shot benchmarks, we use the open-source `lm-evaluation-harness` to standardize prompting and scoring under its default configuration (Gao et al., 2024). Router consistency is quantified by the *Match Score* defined in Appendix C.

### 4.2 Results

**Main Results.** Tables 2 and 3 summarize PTQ performance on OLMoE, DeepSeek-MoE, and Qwen3-MoE. Across both bit settings, RouteQuant attains the best average accuracy among PTQ baselines and closes the perplexity gap to full precision. Relative to DuQuant, RouteQuant improves the average zero-shot accuracy at W4A4 by +4.48% on OLMoE, +3.57% on DeepSeek-MoE, and +1.61% on Qwen3-MoE, and

Table 2: **Main results under W4A4.** DuQuant serves as the baseline; the comparison includes MoEQuant, EAQuant, QESC, and RouteQuant on OLMoE, DeepSeek-MoE, and Qwen3-MoE.

| Model | Method | Perplexity ↓ | | Accuracy ↑ | | | | | | | |
|---|---|---|---|---|---|---|---|---|---|---|---|
| | | Wiki2 | C4 | ARC-C | ARC-E | BoolQ | HellaS | OBQA | RTE | Wino. | Average |
| OLMoE | FP16 | 6.65 | 10.86 | 46.59 | 77.10 | 70.09 | 58.47 | 32.60 | 71.12 | 68.51 | 60.64 |
| | DuQuant | 8.28 | 12.20 | 40.96 | 72.26 | 63.88 | 54.99 | 30.20 | 60.43 | 64.72 | 55.35 (-) |
| | MoEQuant | 8.03 | 12.02 | 40.70 | 73.44 | 66.30 | 55.69 | 29.60 | 62.09 | 63.38 | 55.89 (+0.97%) |
| | EAQuant | 7.75 | 11.75 | 41.38 | **73.65** | **66.97** | 56.05 | 31.00 | 62.45 | 65.43 | 56.70 (+2.45%) |
| | QESC | 7.76 | 11.75 | 41.38 | 73.53 | 65.66 | 55.93 | 29.60 | 65.70 | **65.75** | 56.79 (+2.61%) |
| | RouteQuant | **7.73** | **11.73** | **43.17** | 72.85 | 65.57 | **56.21** | 32.00 | 69.31 | 65.67 | **57.83 (+4.48%)** |
| DeepSeek | FP16 | 6.51 | 9.10 | 45.14 | 75.88 | 72.69 | 58.10 | 32.40 | 62.82 | 70.32 | 59.62 |
| | DuQuant | 7.66 | 10.52 | 39.25 | 71.33 | 64.89 | 53.54 | 25.00 | 57.04 | 64.64 | 53.67 (-) |
| | MoEQuant | 7.55 | 10.24 | 39.08 | 70.83 | 66.67 | 53.54 | 26.60 | 58.24 | 64.09 | 54.15 (+0.89%) |
| | EAQuant | 7.34 | 10.07 | 38.40 | 71.00 | 68.44 | 54.68 | 26.80 | 57.04 | **65.82** | 54.60 (+1.73%) |
| | QESC | 7.35 | 10.10 | 39.58 | 70.83 | 67.22 | 54.58 | 28.40 | 58.24 | 64.25 | 54.73 (+1.97%) |
| | RouteQuant | **7.31** | **10.06** | **40.19** | **72.01** | **68.65** | **55.12** | **28.80** | **59.93** | 64.40 | **55.59 (+3.57%)** |
| Qwen3 | FP16 | 8.70 | 12.31 | 52.56 | 79.34 | 88.75 | 59.52 | 34.00 | 83.03 | 70.32 | 66.79 |
| | DuQuant | 9.86 | 13.62 | 46.33 | 73.44 | **86.67** | 56.71 | **32.60** | **80.51** | 64.25 | 62.93 (-) |
| | MoEQuant | 9.85 | 13.58 | 47.18 | 73.99 | 86.64 | 55.29 | 31.80 | 80.14 | 65.82 | 62.98 (+0.08%) |
| | EAQuant | 9.59 | 13.29 | 49.15 | 75.29 | 86.48 | **56.72** | 30.40 | 77.62 | 66.46 | 63.16 (+0.37%) |
| | QESC | 9.59 | 13.30 | 48.29 | 75.42 | 86.51 | 56.63 | 31.60 | 79.06 | 65.67 | 63.31 (+0.61%) |
| | RouteQuant | **9.58** | **13.28** | **49.66** | **75.84** | 86.64 | 56.69 | 32.40 | 79.78 | **66.61** | **63.95 (+1.61%)** |

Table 3: **Main results under W4A8.** DuQuant serves as the baseline; the comparison includes MoEQuant, EAQuant, QESC, and RouteQuant on OLMoE, DeepSeek-MoE, and Qwen3-MoE.

| Model | Method | Perplexity ↓ | | Accuracy ↑ | | | | | | | |
|---|---|---|---|---|---|---|---|---|---|---|---|
| | | Wiki2 | C4 | ARC-C | ARC-E | BoolQ | HellaS | OBQA | RTE | Wino. | Average |
| OLMoE | FP16 | 6.65 | 10.86 | 46.59 | 77.10 | 70.09 | 58.47 | 32.60 | 71.12 | 68.51 | 60.64 |
| | DuQuant | 7.30 | 11.41 | 43.60 | 73.53 | 65.93 | 56.98 | 31.20 | 62.09 | 66.54 | 57.12 (-) |
| | MoEQuant | 7.22 | 11.30 | **43.94** | **75.21** | 65.47 | 56.52 | 31.00 | 66.45 | 66.85 | 57.92 (+1.39%) |
| | EAQuant | 7.27 | **11.25** | 41.78 | 73.24 | 67.22 | 56.52 | **31.80** | 67.51 | 66.92 | 57.86 (+1.28%) |
| | QESC | 7.21 | 11.26 | 42.66 | 73.53 | **67.74** | 56.61 | 30.80 | 67.23 | 66.77 | 57.91 (+1.37%) |
| | RouteQuant | **7.14** | **11.25** | 41.89 | 72.85 | 67.68 | **57.16** | 31.40 | **68.59** | 67.09 | **58.09 (+1.70%)** |
| DeepSeek | FP16 | 6.51 | 9.10 | 45.14 | 75.88 | 72.69 | 58.10 | 32.40 | 62.82 | 70.32 | 59.62 |
| | DuQuant | 7.02 | 9.70 | 41.47 | 73.44 | 70.37 | 56.21 | **29.20** | 60.28 | 67.32 | 56.90 (-) |
| | MoEQuant | 6.90 | 9.58 | 41.47 | 73.19 | 72.08 | **56.75** | 28.60 | 59.21 | 67.96 | 57.04 (+0.24%) |
| | EAQuant | 6.89 | 9.58 | 41.89 | 74.07 | 71.38 | 56.34 | **29.20** | 60.29 | 67.96 | 57.30 (+0.71%) |
| | QESC | 6.89 | 9.57 | 41.81 | 74.24 | 71.50 | 56.46 | 29.00 | 59.93 | 67.25 | 57.15 (+0.48%) |
| | RouteQuant | **6.88** | **9.56** | **42.83** | **74.28** | **73.12** | 56.68 | **29.20** | 61.37 | **68.27** | **57.96 (+1.87%)** |
| Qwen3 | FP16 | 8.70 | 12.31 | 52.56 | 79.34 | 88.75 | 59.52 | 34.00 | 83.03 | 70.32 | 66.79 |
| | DuQuant | 9.20 | 12.69 | 52.13 | **79.42** | 86.35 | 58.07 | 33.40 | 79.03 | 66.98 | 65.05 (-) |
| | MoEQuant | 9.19 | 12.67 | 51.28 | 77.78 | **88.87** | 58.17 | 33.80 | 80.51 | 67.88 | 65.47 (+0.64%) |
| | EAQuant | 9.13 | 12.62 | 51.54 | 78.20 | 85.85 | 58.52 | 35.00 | 80.87 | 67.32 | 65.33 (+0.42%) |
| | QESC | 9.13 | 12.62 | 50.85 | 78.28 | 86.97 | **58.68** | 33.80 | 80.51 | 67.01 | 65.16 (+0.16%) |
| | RouteQuant | **9.01** | **12.61** | **52.99** | 78.54 | 87.95 | 58.53 | **35.40** | 82.67 | 69.14 | **66.46 (+2.16%)** |

at W4A8 by +1.70% on OLMoE, +1.87% on DeepSeek-MoE, and +2.16% on Qwen3-MoE. It also yields the lowest perplexity among quantized methods under W4A4 and W4A8. Overall, RouteQuant consistently strengthens accuracy while preserving perplexity across diverse MoE backbones and bit budgets, supporting our design of expert-specific scaling and router-consistency losses for robust PTQ of MoE. We also provide the W3A8 experiments in Appendix F.

**Router Consistency (Match Score).** Table 4 reports the agreement between FP16 routing and each PTQ method, while the layer-averaged confusion matrices in Figure 4 visualize the same trend. Across OLMoE, DeepSeek-MoE, and Qwen3-MoE, RouteQuant achieves the highest match scores under both W4A4 and W4A8, indicating that its quantized router decisions remain closest to the full-precision baseline. These

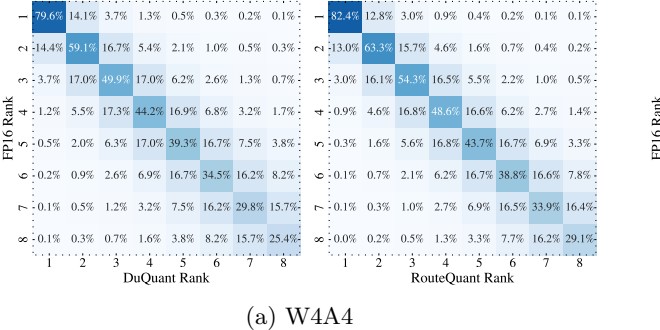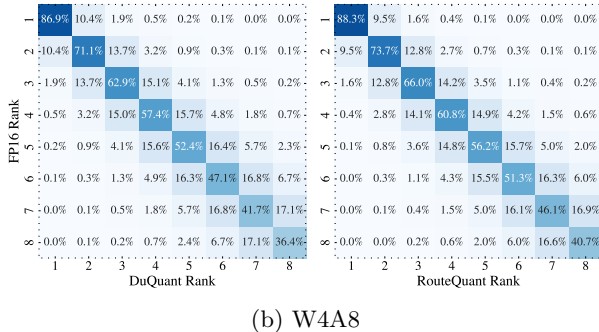

(a) W4A4

(b) W4A8

Figure 4: Layer-averaged confusion matrices of router top-$k$ indices versus FP16 on OLMoE; RouteQuant shows stronger on-diagonal alignment than DuQuant.

Table 4: Match Score under W4A4 and W4A8.

| Bits | Method | Match Score ↑ | | |
|------|--------|-------|----------|-------|
| | | OLMoE | DeepSeek | Qwen3 |
| **W4A4** | DuQuant | 63.71 | 56.11 | 51.68 |
| | MoEQuant | 64.25 | 57.38 | 52.23 |
| | EAQuant | 65.02 | 58.11 | 52.45 |
| | QESC | 65.10 | 58.32 | 52.69 |
| | RouteQuant | **66.95** | **59.24** | **53.97** |
| **W4A8** | DuQuant | 72.87 | 62.56 | 66.17 |
| | MoEQuant | 73.69 | 62.88 | 66.97 |
| | EAQuant | 73.54 | 63.96 | 66.86 |
| | QESC | 73.67 | 63.78 | 66.45 |
| | RouteQuant | **75.25** | **65.00** | **68.13** |

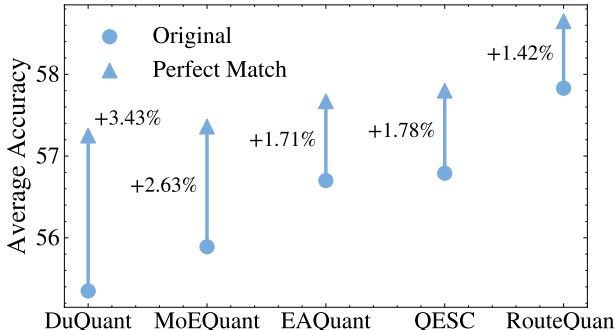

Figure 5: Average accuracy with "Perfect Match" routing, where the selections follows the FP16 top-$k$.

improvements in routing consistency directly parallel the accuracy gains reported in Tables 2 and 3: methods that better preserve router behavior deliver stronger accuracy at the same level. The magnitude of the downstream gain, however, depends on how much of the quantization-induced degradation is attributable to correctable router–expert mismatch. For example, the smaller Match Score improvement on Qwen3-MoE suggests that its baseline routing decisions are already relatively stable, or that the remaining error is dominated more by expert-side quantization noise. This reinforces our design principle that stabilizing router rankings is essential for low-bit MoE inference and translates into downstream performance improvements.

## 4.3 Ablation Study

**Perfect-Match Routing.** To isolate the effect of routing on downstream accuracy, the quantized model is executed under a *perfect-match* scheme in which the selected top-$k$ experts at every token exactly follow the FP16 router while all experts remain quantized. On OLMoE at W4A4, this scheme boosts average accuracy across all methods, validating that routing mismatches are a principal source of PTQ degradation (see Figure 5). The smaller headroom for RouteQuant is consistent with its higher match score, showing that *RAJ* and *GH* already recover much of the routing fidelity.

**Objective-wise Impact.** We quantify the contribution of each objective on OLMoE at W4A4 (see Table 5). Starting from the DuQuant baseline, *ES* alone raises the average to 56.96. On top of *ES*, adding *RAJ* improves performance to 57.25, and *GH* further to 57.62. Applying *RAJ+GH* without *ES* also yields improvement, showing that these objectives are effective on their own. Enabling all three delivers the best result, reinforcing the effectiveness of our design objective.

**Rank-Decay Sensitivity ($\beta$).** We study the weighting parameter $\beta$ in *RAJ* (see Figure 6), which controls how importance decays across expert ranks: $\beta = 1$ assigns a uniform weight to all top-$k$ ranks, while a smaller

Table 5: Ablation of RouteQuant objectives on OLMoE (W4A4).

| ES | RAJ | GH | ARC-C | ARC-E | BoolQ | HellaS | OBQA | RTE | Wino. | Avg. ↑ |
|----|-----|-----|-------|-------|-------|--------|------|-----|-------|--------|
| | | | 40.96 | 72.26 | 63.88 | 54.99 | 30.20 | 60.43 | 64.72 | 55.35 (-) |
| ✓ | | | 41.30 | 73.36 | **65.96** | **56.40** | 31.00 | 64.62 | 66.06 | 56.96 (+2.91%) |
| | ✓ | ✓ | 42.41 | **74.24** | 65.78 | 55.77 | 29.20 | 63.54 | **67.09** | 56.86 (+2.73%) |
| ✓ | ✓ | | 41.89 | 72.84 | 65.38 | 56.25 | 31.00 | 67.51 | 65.87 | 57.25 (+3.43%) |
| ✓ | | ✓ | 43.09 | 73.32 | 65.42 | 56.15 | 31.60 | 68.23 | 65.52 | 57.62 (+4.10%) |
| ✓ | ✓ | ✓ | **43.17** | 72.85 | 65.57 | 56.21 | **32.00** | **69.31** | 65.67 | **57.83** (+4.48%) |

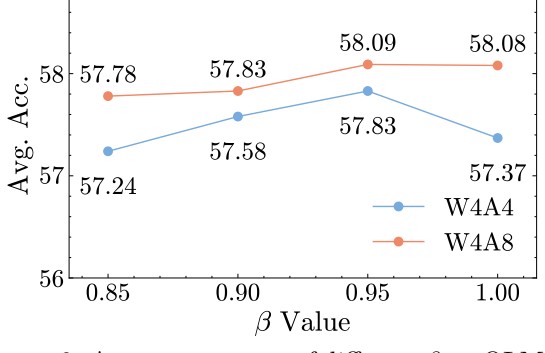

Figure 6: Average accuracy of different $\beta$ on OLMoE under W4A4 and W4A8.

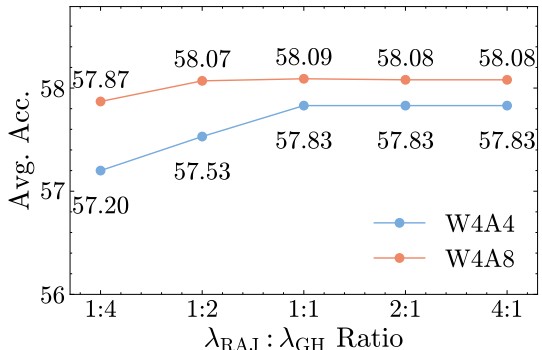

Figure 7: Effect of $\lambda_{\text{RAJ}}{:}\lambda_{\text{GH}}$ on OLMoE under W4A4 and W4A8.

$\beta$ applies an exponential decay that emphasizes higher-ranked experts. Sweeping $\beta \in \{0.85, 0.90, 0.95, 1.00\}$, we find that $\beta = 0.95$ consistently yields the best average accuracy under both W4A4 and W4A8. $\beta \leq 0.90$ overemphasize the top-1 and top-2 experts and amplify noise from minor rank flips, whereas $\beta = 1.00$ spreads weight too uniformly and under-penalizes lower-rank mismatches. The peak at $\beta = 0.95$ indicates that a mild exponential decay strikes the right balance, prioritizing the top of the FP16 ranking while still regularizing lower ranks, thereby improving router consistency and downstream accuracy.

**Balancing Rank and Margin.** The router-loss $\mathcal{L}_{\text{router}}$ (see Figure 7) is controlled by $\lambda_{\text{RAJ}}$ and $\lambda_{\text{GH}}$ to balance between rank and margin preservation. A sweep over ratios {1:4, 1:2, 1:1, 2:1, 4:1} on OLMoE shows that 1:1 attains the highest average accuracy under W4A4 and W4A8. Skewing the weights toward *RAJ* (e.g., 4:1) over-penalizes benign rank shuffles and dampens margins, while favoring *GH* (e.g., 1:4) prioritizes separation but becomes less sensitive to near ties within the top-$k$. The symmetric setting aligns the objectives, improving match score and yielding the best downstream accuracy at the same bit budget.

**Router-Quantized Robustness.** In the main experiments, the router is kept at W8A8 while only the expert modules are quantized. To further validate the robustness of our design, we additionally quantize the router to W4A8, matching the precision used for experts. This setting is substantially more challenging since router logits directly determine expert selection and are highly sensitive to quantization noise. As shown in Table 6, RouteQuant consistently achieves the best average accuracy across OLMoE, DeepSeek-MoE, and Qwen3-MoE. These results further confirm our hypothesis that improving routing stability is crucial under aggressive precision constraints, making RouteQuant more resilient when both routing and expert computation are quantized.

**Runtime Efficiency.** We evaluate decoding throughput (Tokens/Sec.) on a single NVIDIA A100 using prompts of 1024 tokens and generating 128 tokens, and we report the mean ± std over five runs. Table 7 shows that DuQuant, MoEQuant, EAQuant and QESC achieve similar runtimes with small standard deviations, indicating that RouteQuant remains comparable even when incorporating heterogeneous expert scaling. This parity is expected because *ES* introduces only a lightweight per-expert scaling that is applied once during quantization and reduces to a constant multiplication at inference, incurring negligible overhead.

Table 6: **W4A8 results with router at W4A8.** DuQuant serves as the baseline; the comparison includes MoEQuant, EAQuant, QESC, and RouteQuant on OLMoE, DeepSeek-MoE, and Qwen3-MoE.

| Model | Method | Perplexity ↓ | | Accuracy ↑ | | | | | | | |
|---|---|---|---|---|---|---|---|---|---|---|---|
| | | Wiki2 | C4 | ARC-C | ARC-E | BoolQ | HellaS | OBQA | RTE | Wino. | Average |
| OLMoE | FP16 | 6.65 | 10.86 | 46.59 | 77.10 | 70.09 | 58.47 | 32.60 | 71.12 | 68.51 | 60.64 |
| | DuQuant | 7.43 | 11.52 | **42.75** | **74.24** | 65.11 | 56.05 | 29.20 | 62.82 | 66.22 | 56.63 (-) |
| | MoEQuant | 7.20 | 11.32 | 40.36 | 72.26 | **67.74** | 57.20 | 30.80 | 66.45 | **67.80** | 57.52 (+1.57%) |
| | EAQuant | 7.18 | 11.29 | 40.19 | 72.31 | **67.74** | **57.43** | 31.00 | 67.51 | 67.48 | 57.67 (+1.83%) |
| | QESC | 7.18 | 11.30 | 41.47 | 72.90 | 67.22 | 57.35 | 30.80 | 66.15 | 66.69 | 57.51 (+1.56%) |
| | RouteQuant | **7.17** | **11.28** | 42.66 | 73.61 | 66.88 | 57.13 | **31.20** | 68.95 | 66.61 | **58.15 (+2.69%)** |
| DeepSeek | FP16 | 6.51 | 9.10 | 45.14 | 75.88 | 72.69 | 58.10 | 32.40 | 62.82 | 70.32 | 59.62 |
| | DuQuant | 7.03 | 9.74 | 41.81 | 73.27 | 71.74 | **56.63** | 26.80 | 57.76 | **68.03** | 56.58 (-) |
| | MoEQuant | 6.92 | 9.61 | 41.89 | 73.40 | 71.77 | 56.52 | 27.40 | 58.84 | 67.48 | 56.76 (+0.32%) |
| | EAQuant | 6.91 | 9.59 | 41.64 | 74.45 | 71.44 | 56.13 | **29.40** | 59.57 | 66.54 | 57.02 (+0.79%) |
| | QESC | 6.91 | 9.59 | 41.47 | 74.28 | 71.10 | 56.28 | 29.20 | **60.29** | 66.93 | 57.08 (+0.89%) |
| | RouteQuant | **6.90** | **9.58** | **42.83** | **74.79** | **73.06** | 56.62 | 28.80 | 59.57 | 67.01 | **57.53 (+1.68%)** |
| Qwen3 | FP16 | 8.70 | 12.31 | 52.56 | 79.34 | 88.75 | 59.52 | 34.00 | 83.03 | 70.32 | 66.79 |
| | DuQuant | 9.29 | 12.83 | 51.19 | **79.88** | 87.49 | 57.61 | 33.00 | 79.42 | 68.03 | 65.23 (-) |
| | MoEQuant | 9.25 | 12.70 | 52.05 | 77.74 | 88.26 | 58.03 | 34.40 | 80.87 | 67.88 | 65.60 (+0.57%) |
| | EAQuant | 9.31 | 12.71 | 51.11 | 78.54 | 86.70 | **58.36** | 33.40 | 81.59 | 68.67 | 65.48 (+0.38%) |
| | QESC | 9.26 | 12.68 | 51.54 | 79.04 | 86.88 | 58.14 | **34.80** | 79.78 | 68.75 | 65.56 (+0.51%) |
| | RouteQuant | **9.04** | **12.65** | **52.99** | 79.12 | **88.53** | 58.32 | 34.40 | **82.67** | **70.48** | **66.64 (+2.17%)** |

Table 7: Tokens/sec throughput under W4A8. Values are reported as mean±std over 5 runs.

Table 8: Comparison of router precision and RAJ/GH settings under W4A4 (non-router).

| Model | Tokens-per-second ↑ | | |
|---|---|---|---|
| | OLMoE | DeepSeek | Qwen3 |
| DuQuant | 6.52 ± 0.01 | 23.88 ± 0.05 | 1.17 ± 0.01 |
| MoEQuant | 6.84 ± 0.03 | 24.56 ± 0.06 | 1.21 ± 0.02 |
| EAQuant | 6.72 ± 0.05 | 24.14 ± 0.07 | 1.20 ± 0.01 |
| QESC | 6.70 ± 0.03 | 24.08 ± 0.05 | 1.19 ± 0.01 |
| RouteQuant | 6.49 ± 0.03 | 23.79 ± 0.06 | 1.16 ± 0.01 |

| Model | Router | ES | RAJ/GH | Avg. ↑ |
|---|---|---|---|---|
| OLMoE | FP16 | v | x | 57.41 |
| | W4A8/W8A8 | v | x | 56.88/56.96 |
| | W4A8/W8A8 | v | v | 57.73/57.83 |
| DeepSeek | FP16 | v | x | 55.23 |
| | W4A8/W8A8 | v | x | 54.84/54.95 |
| | W4A8/W8A8 | v | v | 55.50/55.59 |
| Qwen3 | FP16 | v | x | 63.52 |
| | W4A8/ W8A8 | v | x | 63.51/63.58 |
| | W4A8/W8A8 | v | v | 63.87/63.95 |

Similarly, *RAJ* and *GH* are optimization objectives that are invoked only in the calibration stage to adjust quantization parameters, and thus do not alter the forward pass or add runtime cost. As a result, the inference computation graph and memory traffic remain identical to the baseline, ensuring that RouteQuant preserves runtime efficiency while providing superior accuracy and perplexity.

**Full-Precision Router in RouteQuant.** We further examine whether keeping the router in full precision is preferable within RouteQuant. To isolate the effect of router precision, we conduct a controlled ablation with fixed expert quantization, calibration data, and *ES* configuration. We compare three variants: an FP16 router with *ES* only, a W8A8 router with *ES* only, and a W8A8 router with *ES+RAJ+GH*, corresponding to full RouteQuant. As shown in Table 8, router quantization alone does not consistently improve performance over the FP16-router setting. However, adding *RAJ* and *GH* to the quantized router consistently achieves the best results across all models. This indicates that RouteQuant's gain comes from combining router quantization with routing-consistency regularization, rather than router quantization alone. It also shows that simply keeping the router in full precision is suboptimal, since it cannot benefit from the proposed router-alignment objectives.

**Weight-only quantization (W4A16).** The main experiments primarily focus on scenarios where both weights and activations are quantized. To assess whether our method remains effective without activation quantization, we further conduct evaluations under the weight-only setting. Compared against strong baselines such as AWQ (Lin et al., 2024b) and GPTQ (Frantar et al., 2022), our method consistently achieves

Table 9: Zero-shot accuracy under W4A16 quantization.

| Model | Method | Accuracy ↑ | | | | | | | |
|---|---|---|---|---|---|---|---|---|---|
| | | ARC-C | ARC-E | BoolQ | HellaS | OBQA | RTE | Wino. | Average |
| OLMoE | FP16 | 46.59 | 77.10 | 70.09 | 58.47 | 32.60 | 71.12 | 68.51 | 60.64 |
| | AWQ | 41.55 | 74.49 | 66.67 | 56.69 | 31.40 | **63.90** | 65.43 | 57.16 |
| | GPTQ | 42.66 | **75.21** | 67.34 | 57.33 | 31.80 | 62.82 | 67.72 | 57.84 |
| | RouteQuant | **45.22** | **75.21** | **70.00** | **57.66** | **32.60** | 62.82 | **68.27** | **58.83** |
| DeepSeek | FP16 | 45.14 | 75.88 | 72.69 | 58.10 | 32.40 | 62.82 | 70.32 | 59.62 |
| | AWQ | 41.38 | 72.22 | 65.66 | 55.73 | 28.40 | 57.04 | 66.61 | 55.29 |
| | GPTQ | 42.49 | 73.61 | 67.55 | 56.21 | 29.40 | 60.43 | 65.67 | 56.48 |
| | RouteQuant | **44.45** | **74.37** | **73.82** | **56.62** | **30.00** | **64.62** | **68.75** | **58.95** |
| Qwen3 | FP16 | 52.56 | 79.34 | 88.75 | 59.52 | 34.00 | 83.03 | 70.32 | 66.79 |
| | AWQ | 45.73 | 73.53 | 86.48 | 56.51 | **33.60** | 80.51 | 67.80 | 63.45 |
| | GPTQ | 48.81 | 76.94 | 87.95 | 57.01 | 32.80 | 77.98 | 68.82 | 64.33 |
| | RouteQuant | **53.07** | **79.25** | **88.69** | **58.66** | 32.80 | **81.23** | **68.98** | **66.10** |

Table 10: Results on VLM benchmarks on Qwen3-VL-MoE (W4A4).

| Method | Accuracy ↑ | | | | | | | | |
|---|---|---|---|---|---|---|---|---|---|
| | GQA | ChartQA | ScienceQA | RWQA | K-DTC | OCR | MMMU | ai2d | Average |
| FP16 | 64.04 | 85.28 | 93.63 | 66.14 | 83.33 | 84.80 | 52.00 | 86.08 | 76.91 |
| DuQuant | 59.42 | 79.52 | 90.71 | 63.22 | 79.21 | 81.28 | 48.19 | 82.36 | 72.99 (-) |
| MoEQuant | 60.26 | 80.28 | 91.93 | 63.22 | 80.55 | 82.31 | 48.22 | **83.97** | 73.84 (+1.17%) |
| EAQuant | 61.92 | 82.44 | 91.56 | 63.82 | 80.73 | 82.58 | 49.13 | 83.16 | 74.42 (+1.96%) |
| QESC | 61.87 | **84.22** | 91.44 | 63.65 | 80.68 | 82.47 | 49.22 | 83.08 | 74.58 (+2.18%) |
| RouteQuant | **62.08** | 84.17 | **92.17** | **64.08** | **81.19** | **82.98** | **50.90** | 83.16 | **75.09 (+2.88%)** |

large gains in zero-shot accuracy (see Table 9). This demonstrates that our approach not only addresses activation outliers but also provides robust improvements when only the weight domain is quantized.

# 5 Extending to Multimodal MoEs

To further demonstrate that our proposed quantization framework generalizes beyond language-only MoE models, we additionally evaluate it on a state-of-the-art multimodal architecture. Specifically, we conduct experiments on the Qwen3-VL-30B-A3B-Instruct model (Yang et al., 2025) to assess whether the same quantization strategies remain effective in vision–language settings.

For post-training calibration, we sample image–text pairs from the Flickr30k dataset (Plummer et al., 2015), following an analogous setup to our experiments. We quantize the model using the same configuration as in prior sections, with per-token activation quantization and per-channel weight quantization, and we evaluate the W4A4 variant. Multimodal performance is assessed using `lmms-eval` under its default evaluation protocol. We test across a wide range of benchmarks covering visual reasoning, document understanding, scientific question answering, and real-world perception: GQA (Hudson & Manning, 2019), ChartQA (Masry et al., 2022), ScienceQA (Lu et al., 2022), RealWorldQA (Zhang et al., 2025), K-DTCBench (Ju et al., 2024), OCRBench (Liu et al., 2024c), MMMU (Yue et al., 2023), and AI2D (Kembhavi et al., 2016).

Across all datasets, we observe that the W4A4 version of our method consistently outperforms competing quantization baselines (see Table 10). Notably, the gap is especially clear on multimodal reasoning and knowledge-intensive tasks. These findings validate that the proposed approach is not only effective for large MoE language models but also transfers robustly to more complex multimodal systems, highlighting its versatility and practical utility across diverse architectures.

# 6 Conclusion and Future Work

This paper studies PTQ of MoE models and establishes that router performance governs low-bit accuracy, with errors concentrating as near-neighbor flips near the top-$k$ experts and arising from small score margins. We present RouteQuant, a training-free, calibration-only framework that combines *ES* with *RAJ* and *GH* objectives to stabilize expert rankings and preserve margins; empirically, the framework yields lower perplexity on C4 and WikiText-2 and higher zero-shot accuracy on OLMoE, DeepSeek-MoE, and Qwen3-MoE under W4A4 and W4A8. Looking ahead, we plan to extend our framework to multimodal MoE settings, where heterogeneous modalities may amplify router instability under quantization. We also aim to study adaptive precision guided by observed rank gaps and load patterns, to further enhance robustness and generalization across diverse tasks.

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

# A    Theoretical Analysis of Router Quantization

In this section we provide proofs for Proposition 1: *Router-induced MoE output error* and Proposition 2: *Gap condition for top-k stability.*

## A.1    Proof of Proposition 1

*Proof.* For a fixed input $x$, we have:

$$y^{(\text{fp})}(x) = \sum_{j \in I} \pi_j^{(\text{fp})}(x)\, \mathcal{E}_j, \tag{18}$$

$$y^{(q)}(x) = \sum_{j \in J} \pi_j^{(q)}(x)\, \mathcal{E}_j, \tag{19}$$

where we omit $x$ for brevity.

Using the decomposition $I = S^* \cup S^-$ and $J = S^* \cup S^+$, we have:

$$y^{(\text{fp})} - y^{(q)} = \sum_{j \in S^*} \big(\pi_j^{(\text{fp})} - \pi_j^{(q)}\big)\mathcal{E}_j + \sum_{j \in S^-} \pi_j^{(\text{fp})}\mathcal{E}_j - \sum_{j \in S^+} \pi_j^{(q)}\mathcal{E}_j. \tag{20}$$

Taking norms and applying the triangle inequality yields:

$$\big\|y^{(\text{fp})} - y^{(q)}\big\| \leq \sum_{j \in S^*} \big|\pi_j^{(\text{fp})} - \pi_j^{(q)}\big|\, \|\mathcal{E}_j\| + \sum_{j \in S^-} \pi_j^{(\text{fp})}\|\mathcal{E}_j\| + \sum_{j \in S^+} \pi_j^{(q)}\|\mathcal{E}_j\|. \tag{21}$$

Under the assumption that $\|\mathcal{E}_j(x)\| \leq B(x)$ for all $j$, we obtain:

$$\big\|y^{(\text{fp})}(x) - y^{(q)}(x)\big\| \leq B(x)\Big( \sum_{j \in S^*} \big|\pi_j^{(\text{fp})}(x) - \pi_j^{(q)}(x)\big| + \sum_{j \in S^-} \pi_j^{(\text{fp})}(x) + \sum_{j \in S^+} \pi_j^{(q)}(x)\Big), \tag{22}$$

which is exactly Equation (7).    □

## A.2    Proof of Proposition 2

*Proof.* Let $I = (i_1, \ldots, i_k)$ denote the ordered top-$k$ indices under the full-precision logits $r^{(\text{fp})}$, and let $I^c$ be its complement. Define the consecutive gaps along the FP ordering as

$$\Delta_r^{(\text{fp})} = r_{i_r}^{(\text{fp})} - r_{i_{r+1}}^{(\text{fp})} > \epsilon, \quad r = 1, \ldots, k. \tag{23}$$

Assume that the quantized logits satisfy the gap condition along the same FP order:

$$\Delta_r^{(q)} \triangleq r_{i_r}^{(q)} - r_{i_{r+1}}^{(q)} \geq \Delta_r^{(\text{fp})}, \quad r = 1, \ldots, k. \tag{24}$$

Then, for any $r \in \{1, \ldots, k\}$, summing over consecutive gaps yields

$$r_{i_r}^{(q)} - r_{i_{k+1}}^{(q)} = \sum_{t=r}^{k} \Delta_t^{(q)} \geq \sum_{t=r}^{k} \Delta_t^{(\text{fp})} = r_{i_r}^{(\text{fp})} - r_{i_{k+1}}^{(\text{fp})} > 0, \tag{25}$$

which implies $r_{i_r}^{(q)} > r_{i_{k+1}}^{(q)}$ for all $r \leq k$.

Moreover, since $i_{k+1}$ is the highest-ranked index in $I^c$ under the FP ordering, we have $r_{i_{k+1}}^{(q)} \geq r_j^{(q)}$ for all $j \in I^c$ whenever the FP order is preserved on $I^c$. Therefore,

$$r_{i_r}^{(q)} > r_j^{(q)}, \quad \forall r \leq k, \ \forall j \in I^c, \tag{26}$$

which shows that no index in $I^c$ can enter the top-$k$ under quantization. Hence, the ordered top-$k$ expert indices are preserved with full precision.    □

Table 11: Performance on math and coding benchmarks.

| Method | GSM8K ↑ (8-shot CoT EM) | HumanEval ↑ (0-shot Pass@1) |
|---|---|---|
| FP16 | 87.89 | 41.70 |
| DuQuant | 75.72 | 32.34 |
| MoEQuant | 77.12 | 35.83 |
| EAQuant | 78.25 | 37.20 |
| QESC | 78.33 | 36.89 |
| RouteQuant | **80.21** | **38.41** |

## B  Math and Coding Evaluation

In addition to the main benchmarks, we further evaluate our method on representative math and coding tasks to assess its robustness on reasoning-intensive workloads. Specifically, we report results on GSM8K (8-shot chain-of-thought exact match) for mathematical reasoning (Cobbe et al., 2021) and HumanEval (0-shot Pass@1) for code generation (Chen et al., 2021). As shown in Table 11, while post-training quantization leads to noticeable performance degradation compared to FP16, RouteQuant consistently outperforms other MoE quantization baselines on both tasks. These results suggest that preserving routing consistency not only benefits language modeling and multiple-choice benchmarks, but also improves robustness on complex reasoning and program synthesis tasks.

## C  Match Score for Router Consistency

We quantify the agreement between FP16 and quantized routing with a *Match Score* that evaluates whether the quantized router preserves both the membership and the order of the FP16 top-$k$ experts. For each MoE layer $\ell \in \{1, \dots, L\}$, let:

$$I^{(\ell)} = (i_1^{(\ell)}, \dots, i_k^{(\ell)}) = \text{top-}k(\mathbf{r}_\ell^{(\text{fp})}, k),$$

$$J^{(\ell)} = (j_1^{(\ell)}, \dots, j_k^{(\ell)}) = \text{top-}k(\mathbf{r}_\ell^{(\text{q})}, k),$$

denote the ordered top-$k$ expert indices from FP16 and quantized router logits, respectively, where top-$k(\cdot, k)$ returns the indices sorted by descending scores. Define the quantized rank function:

$$\rho_\ell^{(\text{q})}(e) = \begin{cases} s, & \text{if } e = j_s^{(\ell)} \text{ with } s \in \{1, \dots, k\}, \\ \infty, & \text{if } e \notin J^{(\ell)}, \end{cases}$$

so that an FP16-selected expert missing from the quantized top-$k$ receives infinite rank (and thus zero credit). The layerwise agreement is then averaged over the $k$ FP16-selected experts and over all $L$ layers:

$$S = \frac{1}{kL} \sum_{\ell=1}^{L} \sum_{r=1}^{k} \frac{1}{1 + |r - \rho_\ell^{(\text{q})}(i_r^{(\ell)})|}, \tag{27}$$

with the convention $1/(1 + \infty) = 0$. By construction, $S \in [0, 1]$; $S = 1$ if and only if the quantized top-$k$ exactly matches FP16, including ranks in every layer, and $S$ approaches 0 when FP16-selected experts are consistently absent from the quantized top-$k$ or appear only at much lower ranks. We compute $\mathbf{r}_\ell^{(\text{fp})}$ and $\mathbf{r}_\ell^{(\text{q})}$ for each input token and then report the dataset-level Match Score by averaging Equation (27) over tokens in a held-out set. Equivalently, if $S^{(t)}$ denotes Equation (27) evaluated on token $t$, the reported score is:

$$\bar{S} = \frac{1}{T} \sum_{t=1}^{T} S^{(t)},$$

where $T$ is the number of evaluated tokens. This metric is sensitive to both rank inversions within the selected set and omissions of FP16-selected experts, which makes it a faithful proxy for router consistency under quantization.

Table 12: Comparison between RouteQuant and existing MoE PTQ methods. "△" indicates that the method partially considers the corresponding factor.

| Method | Q-Router | Logit Align. | Top-$k$ Align. | Rank | Gap | ES | MM Eval. |
|---|---|---|---|---|---|---|---|
| MoEQuant | ✗ | ✗ | ✗ | ✗ | ✗ | ✗ | ✗ |
| QESC | ✗ | ✓ | △ | ✗ | ✗ | ✗ | ✗ |
| EAQuant | ✓ | ✓ | △ | ✗ | ✗ | △ | ✗ |
| RouteQuant | ✓ | ✓ | ✓ | ✓ | ✓ | ✓ | ✓ |

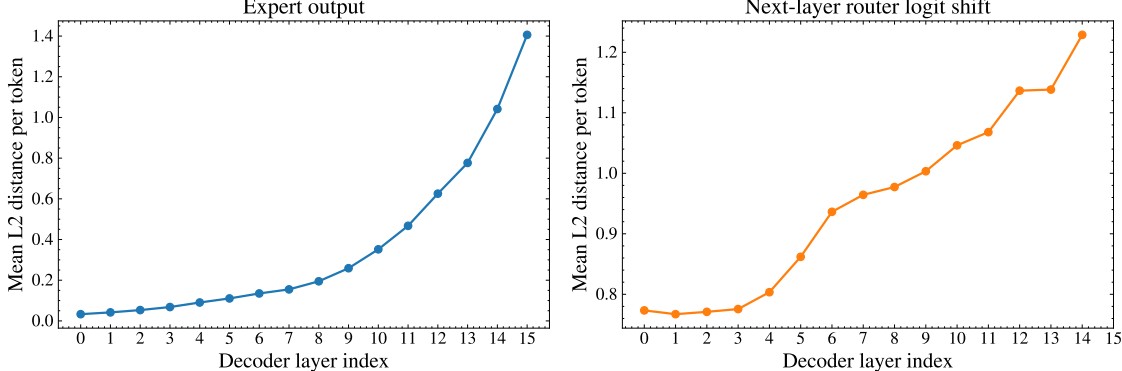

Figure 8: Per-layer L2 distance between FP16 and quantized expert outputs on OLMoE under W4A4.

# D  Comparison with Existing MoE PTQ Methods

To better clarify the distinction between RouteQuant and prior MoE PTQ methods, we provide a summarized comparison in Table 12. Existing approaches mainly focus on expert-balanced calibration, affinity-aware expert quantization, or router-logit/distribution alignment. In contrast, RouteQuant explicitly models the two routing factors identified in our theoretical analysis: the top-$k$ expert identity/order through RAJ and the inter-expert logit gaps through GH. In addition, RouteQuant introduces expert-specific smoothing to address expert heterogeneity and evaluates the framework on multimodal MoEs, further distinguishing it from prior MoE PTQ methods.

# E  Direct Evidence of Expert-Output Shift

While Figure 1 provides routing-level evidence, we further add a direct measurement of expert-output shift. Specifically, on OLMoE under W4A4, we compute the L2 distance between FP16 expert outputs and their quantized counterparts at each decoder layer, using the same input tokens and averaging over tokens and selected experts. As shown in Figure 8, the L2 distance is small in early layers but increases steadily with depth, becoming much larger in later MoE layers. This confirms that quantization induces expert-output shifts that accumulate across layers. We also observe a similar increasing trend for the next-layer router logit shift, indicating that these shifted expert outputs propagate to downstream routers and can make a frozen full-precision router mismatched with the quantized expert pathway.

# F  Additional Experiments

**Portability to SmoothQuant.**  To directly examine whether the proposed components are specific to DuQuant Lin et al. (2024a), we conduct an additional experiment by integrating them into SmoothQuant Xiao et al. (2023) on OLMoE under W4A4 quantization. As shown in Table 13, adding ES, RAJ, and GH improves the SmoothQuant baseline from 55.07 to 56.79 average zero-shot accuracy, corresponding to a relative improvement of +3.12%. These results suggest that the proposed components are not tied to DuQuant and can generalize to another calibration-based quantization framework. This portability is

Table 13: Zero-shot accuracy of integrating the proposed components into SmoothQuant on OLMoE under W4A4 quantization.

| Method | ARC-C | ARC-E | BoolQ | HellaS | OBQA | RTE | Wino. | Average |
|---|---|---|---|---|---|---|---|---|
| SmoothQuant | 39.68 | 69.99 | 62.29 | 54.00 | **30.20** | 63.90 | 65.43 | 55.07 (-) |
| SmoothQuant + ES/RAJ/GH | **42.15** | **73.65** | **65.62** | **55.69** | 29.40 | **64.21** | **66.80** | **56.79** (+3.12%) |

Table 14: **Results under W3A8.** DuQuant serves as the baseline; the comparison includes MoEQuant, EAQuant, QESC, and RouteQuant on OLMoE, DeepSeek-MoE, and Qwen3-MoE.

| Model | Method | Perplexity ↓ | | Accuracy ↑ | | | | | | | |
|---|---|---|---|---|---|---|---|---|---|---|---|
| | | Wiki2 | C4 | ARC-C | ARC-E | BoolQ | HellaS | OBQA | RTE | Wino. | Average |
| **OLMoE** | FP16 | 6.65 | 10.86 | 46.59 | 77.10 | 70.09 | 58.47 | 32.60 | 71.12 | 68.51 | 60.64 |
| | DuQuant | 10.78 | 14.33 | 36.00 | 66.71 | 61.74 | 50.76 | 27.60 | **61.37** | 59.12 | 51.90 (-) |
| | MoEQuant | 8.89 | 12.74 | 37.28 | 68.73 | **66.91** | 54.35 | 28.20 | 57.04 | 62.98 | 53.64 (+3.36%) |
| | EAQuant | 8.79 | 12.66 | 37.12 | 68.73 | 62.08 | 54.46 | 28.60 | 59.57 | 64.48 | 53.58 (+3.23%) |
| | QESC | 8.79 | 12.67 | 37.03 | 68.31 | 61.93 | 54.42 | 28.20 | 59.21 | 63.93 | 53.29 (+2.68%) |
| | RouteQuant | **8.75** | **12.65** | **38.48** | **70.79** | 63.85 | **54.58** | **30.60** | 60.65 | **65.75** | **54.96** (+5.89%) |

Table 15: Zero-shot accuracy under W4A4 quantization with **128** calibration samples from **C4**.

| Model | Method | Accuracy ↑ | | | | | | | |
|---|---|---|---|---|---|---|---|---|---|
| | | ARC-C | ARC-E | BoolQ | HellaS | OBQA | RTE | Wino. | Average |
| **OLMoE** | FP16 | 46.59 | 77.10 | 70.09 | 58.47 | 32.60 | 71.12 | 68.51 | 60.64 |
| | DuQuant | **40.53** | 70.24 | 65.20 | 55.20 | 29.80 | 62.82 | 63.06 | 55.26 |
| | MoEQuant | 39.85 | **71.59** | 65.81 | 54.86 | 29.60 | 62.82 | **65.43** | 55.71 |
| | EAQuant | 39.24 | 71.00 | **65.66** | 55.01 | 31.00 | 63.54 | 63.93 | 55.63 |
| | QESC | 39.16 | 70.45 | 65.81 | **56.01** | 30.80 | 64.62 | 64.33 | 55.88 |
| | RouteQuant | 39.76 | 71.00 | **65.66** | 55.81 | **31.80** | **66.79** | 64.17 | **56.43** |

possible because the proposed components rely only on quantities generally available during PTQ calibration. Specifically, RAJ and GH require the full-precision and quantized router logits, while ES requires per-expert activation and weight statistics. Therefore, these components can be incorporated into other quantization frameworks that expose quantized router outputs and allow quantization parameters or smoothing factors to be adjusted.

**Lower-Precision Robustness.** Pushing weights to 3-bit while keeping activations at 8-bit stresses the MoE pipeline, yet RouteQuant remains the most robust. All methods experience a drop in accuracy relative to FP16, yet RouteQuant shows the smallest gap and attains the best average accuracy (see Table 14). The trend aligns with our routing hypothesis: lower weight precision amplifies rank flips and narrows score margins, making router consistency increasingly critical. *ES* stabilizes per-expert activation ranges, and *RAJ/GH* improve the match score by preserving rankings and margin gaps, which in turn sustains downstream accuracy under the more aggressive W3A8 setting.

**Varying calibration samples.** Throughout the main experiments, the number of calibration samples is fixed at 256. To examine the sensitivity of our method to this choice, we additionally conduct experiments with 128 and 512 calibration samples (see Table 15 and 16). In both cases, our method continues to outperform all baselines by a clear margin. These results suggest that our method is robust across different calibration budgets and does not rely on a carefully chosen sample size to deliver improvements.

**Alternative calibration dataset.** The primary results in the paper adopt C4 as the calibration dataset. To evaluate whether our method generalizes to other corpora, we also perform calibration on WikiText-2

Table 16: Zero-shot accuracy under W4A4 quantization with **512** calibration samples from **C4**.

| Model | Method | Accuracy ↑ | | | | | | | |
|-------|--------|-------|-------|-------|--------|------|------|-------|---------|
| | | ARC-C | ARC-E | BoolQ | HellaS | OBQA | RTE | Wino. | Average |
| **OLMoE** | FP16 | 46.59 | 77.10 | 70.09 | 58.47 | 32.60 | 71.12 | 68.51 | 60.64 |
| | DuQuant | 38.74 | 69.74 | 65.66 | 55.20 | 29.80 | 63.54 | 63.85 | 55.22 |
| | MoEQuant | 40.53 | 72.10 | 63.67 | 54.96 | 28.40 | 65.70 | 64.56 | 55.70 |
| | EAQuant | 41.72 | 72.85 | 65.90 | 55.92 | 30.60 | 67.51 | **66.56** | 57.29 |
| | QESC | 42.32 | 72.64 | **66.06** | **56.11** | **31.80** | 67.15 | 66.06 | 57.45 |
| | RouteQuant | **42.66** | **74.62** | 65.63 | 55.91 | 31.60 | **69.31** | 66.46 | **58.03** |

Table 17: Zero-shot accuracy under W4A4 quantization with **256** calibration samples from **WikiText-2**.

| Model | Method | Accuracy ↑ | | | | | | | |
|-------|--------|-------|-------|-------|--------|------|------|-------|---------|
| | | ARC-C | ARC-E | BoolQ | HellaS | OBQA | RTE | Wino. | Average |
| **OLMoE** | FP16 | 46.59 | 77.10 | 70.09 | 58.47 | 32.60 | 71.12 | 68.51 | 60.64 |
| | DuQuant | 39.16 | 69.82 | 64.37 | 55.01 | 27.40 | 62.45 | 62.19 | 54.34 |
| | MoEQuant | 39.60 | 72.18 | 67.80 | 55.30 | 28.40 | **63.18** | 65.75 | 56.03 |
| | EAQuant | 41.72 | 73.86 | 65.81 | 55.93 | 28.20 | 62.85 | 64.17 | 56.08 |
| | QESC | 42.24 | **74.58** | 65.26 | 55.83 | 28.20 | 62.85 | 64.96 | 56.27 |
| | RouteQuant | **42.49** | 73.95 | **68.01** | **56.02** | **28.60** | **63.18** | **65.90** | **56.88** |

(see Table 17). Even under this shift in data distribution, our method maintains consistent advantages over all competing approaches. This highlights that our improvements are not tied to a specific calibration corpus and confirms the general applicability of our design.

**Calibration Time Comparison.**  To evaluate the practical calibration overhead of RouteQuant, we measure the wall-clock calibration time on OLMoE using the same setup as our main experiments: a single NVIDIA A100 80GB GPU, 256 C4 calibration sequences, and a sequence length of 2048. We repeat each measurement five times and report the mean and standard deviation. As shown in Table 18, RouteQuant takes 37.34 minutes on average, which is comparable to other router-aware MoE PTQ methods such as EAQuant and QESC. The results confirm that the calibration cost of RouteQuant remains comparable to existing router-aware MoE PTQ methods. Although RouteQuant requires additional calibration compared with the DuQuant backbone, its wall-clock time is close to EAQuant and QESC, indicating that the proposed RAJ/GH objectives do not introduce excessive overhead beyond existing router-aware calibration methods. Moreover, RAJ and GH are used only during calibration and therefore do not affect inference-time computation, while ES is computed once from calibration statistics.

**Controlled Comparison of Router Precision.**  To further clarify the effect of router quantization, we provide a controlled comparison on OLMoE under W4A4 (non-router). We keep the expert quantization setting, calibration data, and ES configuration fixed, and only vary the router precision. As shown in Table 19, W4A8 and W8A8 outperform the full-precision router. This supports our observation that keeping the router frozen in full precision is not always optimal once the experts are quantized. The W8A8 router achieves the best average accuracy, improving over the full-precision router by 0.42 points and over the W4A8 router by 0.10 points. This suggests that the gain does not come simply from perturbing the router, but from allowing it to co-adapt with quantized experts while preserving sufficient routing stability. Overall, this controlled comparison reinforces that router quantization is beneficial when combined with alignment objectives that constrain top-$k$ consistency and margin preservation.

**Robustness Across Calibration Seeds.**  To examine whether the observed improvements depend on a particular calibration sample or random seed, we conduct an additional robustness study on OLMoE under

Table 18: Calibration time comparison on OLMoE. Each method is measured over five runs, and we report the mean and standard deviation.

| Method | Calibration Time (mins) |
|---|---|
| DuQuant | $15.75 \pm 0.32$ |
| MoEQuant | $22.65 \pm 0.46$ |
| EAQuant | $37.71 \pm 0.44$ |
| QESC | $37.58 \pm 0.68$ |
| RouteQuant | $37.34 \pm 0.52$ |

Table 19: Controlled comparison of different router precisions on **OLMoE** under **W4A4** expert quantization. The expert quantization setting, calibration data, and ES configuration are fixed.

| Model | Router | Accuracy ↑ | | | | | | | |
|---|---|---|---|---|---|---|---|---|---|
| | | ARC-C | ARC-E | BoolQ | HellaS | OBQA | RTE | Wino. | Average |
| **OLMoE** | W4A8 | 40.70 | **74.12** | **66.15** | 55.99 | **32.40** | 67.87 | **66.85** | 57.73 |
| | W8A8 | **43.17** | 72.85 | 65.57 | **56.21** | 32.00 | 69.31 | 65.67 | **57.83** |
| | FP16 | 40.27 | 72.43 | 65.38 | 55.96 | 31.60 | **70.42** | 65.82 | 57.41 |

Table 20: Robustness across calibration seeds on **OLMoE** under **W4A4** quantization. We report the mean and standard deviation across five runs.

| Model | Method | Accuracy ↑ | | | | | | | |
|---|---|---|---|---|---|---|---|---|---|
| | | ARC-C | ARC-E | BoolQ | HellaS | OBQA | RTE | Wino. | Average |
| **OLMoE** | FP16 | 46.59 | 77.10 | 70.09 | 58.47 | 32.60 | 71.12 | 68.51 | 60.64 |
| | DuQuant | $40.87 \pm 1.36$ | $72.33 \pm 0.57$ | $63.98 \pm 1.04$ | $54.87 \pm 0.62$ | $30.20 \pm 0.66$ | $60.47 \pm 0.76$ | $64.55 \pm 0.49$ | 55.32 |
| | MoEQuant | $40.66 \pm 0.68$ | $\mathbf{73.58} \pm 1.41$ | $66.29 \pm 0.55$ | $55.77 \pm 0.89$ | $29.80 \pm 0.43$ | $62.21 \pm 1.25$ | $63.56 \pm 0.72$ | 55.98 |
| | EAQuant | $41.45 \pm 1.09$ | $73.52 \pm 0.46$ | $\mathbf{66.86} \pm 1.32$ | $56.11 \pm 0.61$ | $30.60 \pm 0.28$ | $62.28 \pm 0.28$ | $65.38 \pm 1.16$ | 56.60 |
| | QESC | $41.32 \pm 0.44$ | $73.38 \pm 0.74$ | $65.59 \pm 1.21$ | $55.87 \pm 0.44$ | $30.00 \pm 0.76$ | $65.72 \pm 1.03$ | $\mathbf{65.88} \pm 0.52$ | 56.82 |
| | RouteQuant | $\mathbf{42.86} \pm 0.75$ | $73.24 \pm 0.58$ | $66.34 \pm 0.72$ | $\mathbf{56.31} \pm 0.16$ | $\mathbf{32.40} \pm 0.78$ | $\mathbf{68.29} \pm 1.66$ | $65.48 \pm 0.84$ | **57.85** |

W4A4 quantization. We use the same calibration setup as in the main experiments: 256 C4 calibration samples with a sequence length of 2048. For each method, we repeat calibration with five different random seeds and report the mean and standard deviation. As shown in Table 20, RouteQuant consistently outperforms all baselines across seeds, achieving an average accuracy of 57.85, which is very close to the single-seed result in Table 2. It also improves over DuQuant by +4.57% while maintaining small standard deviations across tasks. These results indicate that the gains from ES, RAJ, and GH are stable and reflect systematic improvements in MoE quantization rather than seed-specific effects.

# G  Additional Visualizations

To complement the analyses in the main text, we provide extended visualizations for both router behavior and expert activations. First, confusion matrices for additional OLMoE layers are shown in Figures 9 and 10. These figures confirm our earlier observation that router errors after quantization are highly localized: most mis-selections occur between neighboring experts, with errors clustering near the diagonal rather than spreading arbitrarily. This further supports the claim that router performance is the dominant factor for MoE quantization.

Second, we present detailed activation distributions across experts at different depths. As shown in Figures 11 to 13, experts within the same layer exhibit markedly heterogeneous activation ranges. This reinforces the motivation behind our *Expert-Aware Smoothing Factor*, which assigns each expert its own scaling factor instead of relying on max-aggregation across experts. These visualizations clearly illustrate why per-expert scaling avoids over-clipping light-tailed experts and prevents wasted quantization levels on the majority of channels.

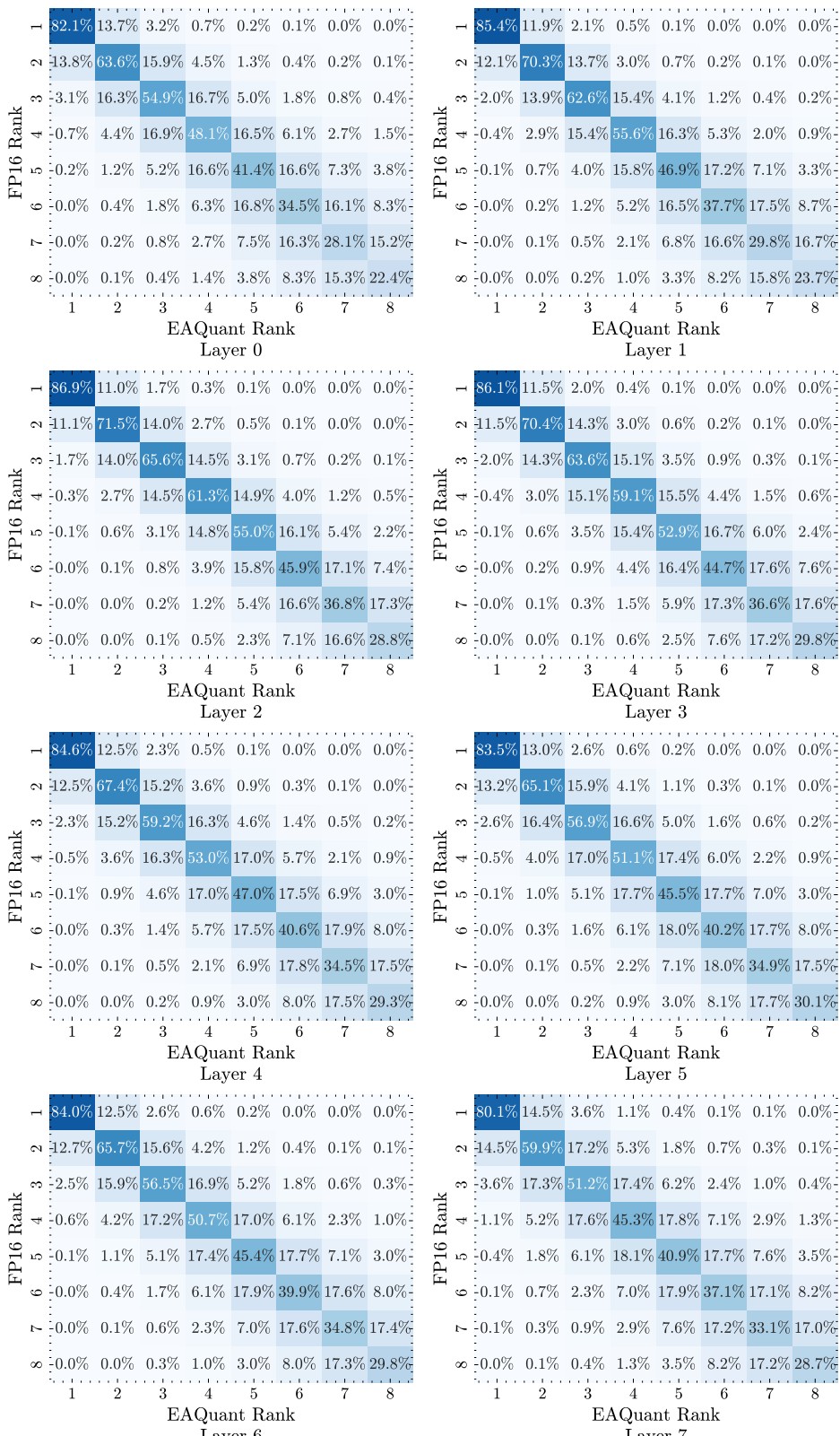

Figure 9: Confusion matrices at **layer 0-7** comparing FP16 vs. EAQuant top-$k$ indices under W4A4.

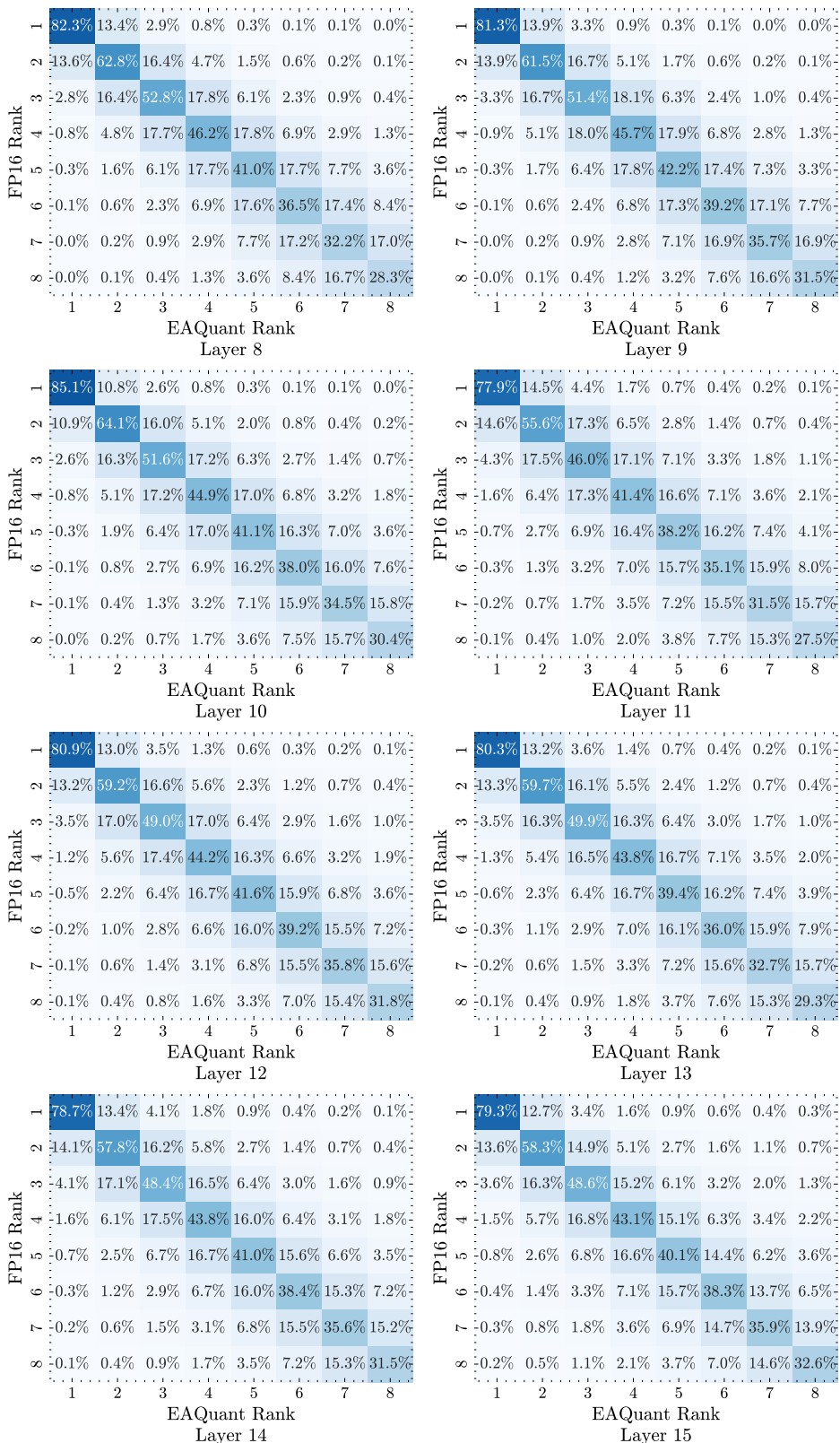

Figure 10: Confusion matrices at **layer 8-15** comparing FP16 vs. EAQuant top-$k$ indices under W4A4.

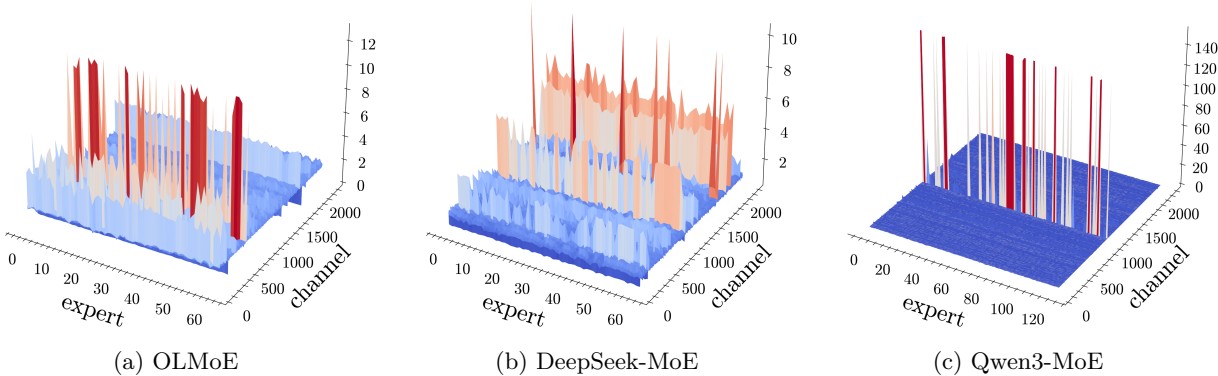

(a) OLMoE        (b) DeepSeek-MoE        (c) Qwen3-MoE

Figure 11: Expert activations in the **2nd layer** for each model.

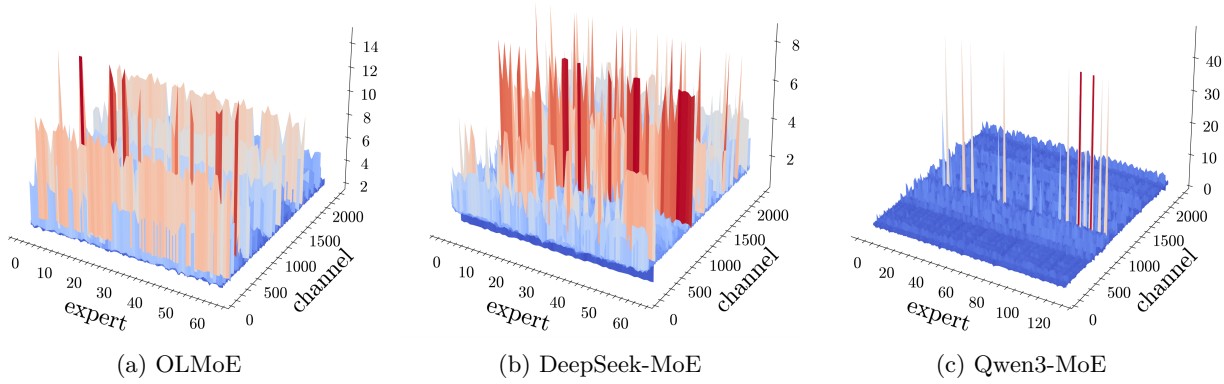

(a) OLMoE        (b) DeepSeek-MoE        (c) Qwen3-MoE

Figure 12: Expert activations in the **6th layer** for each model.

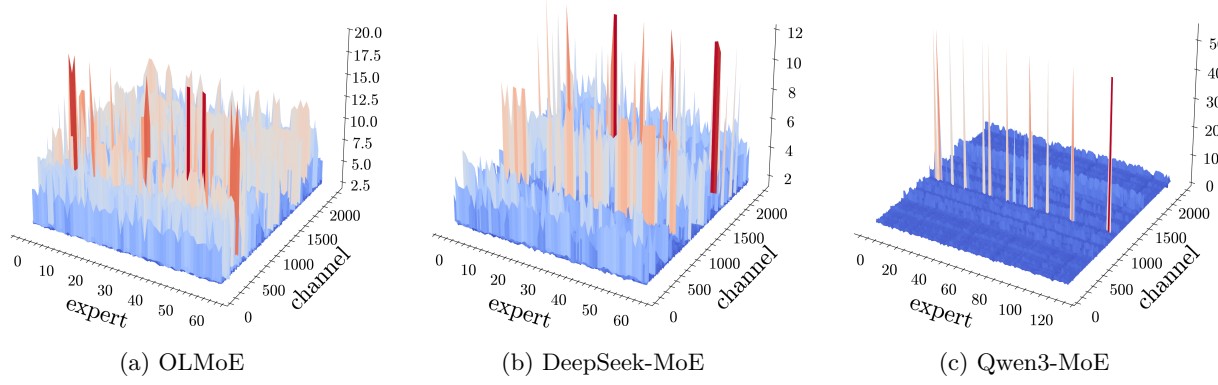

(a) OLMoE        (b) DeepSeek-MoE        (c) Qwen3-MoE

Figure 13: Expert activations in the **10th layer** for each model.

