# OpenReview forum: "Beyond Freezing the Router: Rank-Aligned Post-Training Quantization for Mixture-of-Experts Models"
_TMLR — Accepted by TMLR_

### Review · Reviewer_vudq · 2026-04-28

**Summary Of Contributions:**

This paper focus on post-training quantization for MoE language model. The authors show that a frozen full-precision router cannot compensate for quantization-induced shifts in expert outputs.
They propose RouteQuant, a framework that instead quantizes the router alongside the experts to enable co-adaptation.
Pros:
-  a theoretical analysis (Propositions 1 and 2) decomposing MoE output error under quantization into top-k index mismatch and inter-expert logit gap terms. The analysis is high quality.
- two router-alignment losses and Expert-Aware Smoothing Factor seems reasonable
- the experiment section is solid
- open source

Cons:
- The method is built on top of DuQuant specifically, and it is unclear how portable the approach is to other base quantization frameworks

**Audience:**

Yes

**Audience Explanation:**

MoE architectures are increasingly central to state-of-the-art LLMs

**Claims And Evidence:**

Yes

**Claims Explanation:**

The paper makes 2 main claims, each supported by appropriate evidence:
- Claim 1: Keeping the router in full precision is suboptimal. This is supported by the confusion matrix analysis (Figure 1) and by the empirical comparison in Figure 2. Table 8 further confirms this within the RouteQuant framework itself
- Claim 2: Router-induced error decomposes into top-k index mismatch and inter-expert logit gaps. Propositions 1 and 2 provide formal proofs and the ablation study (Table 5) shows that each component contributes independently, consistent with the theoretical decomposition

**Requested Changes:**

- Generalizability beyond DuQuant. It would be valuable to see a discussion (or ideally a small experiment) on whether RAJ, GH, and ES can be integrated with other base quantization frameworks beyond DuQuant. Even a brief analysis of what assumptions tie the current implementation to DuQuant would help readers assess the broader applicability of the approach.
- Discussion of when gains are smaller. The improvements vary across models. It would be helpful if the authors could provide some intuition for this variation
- Calibration cost. Inference runtime is well-documented (Table 7), but reporting the wall-clock calibration time for RouteQuant compared to baselines would be a useful addition for practitioners.

---

> ### Author Response · Authors · 2026-05-25
> **Response to Reviewer vudq (1/2)**
>
> Thank you for the constructive review and for recognizing the quality of our theoretical analysis, the reasonableness of RAJ/GH and Expert-Aware Smoothing, and the solidity of our experiments. We also appreciate your positive assessment that the main claims are supported by both theoretical and empirical evidence. Your comments on portability beyond DuQuant, model-dependent gain variation, and calibration cost are helpful, and we address them below.
>
> ---
>
> > **Weakness 1**: Generalizability beyond DuQuant. It would be valuable to see a discussion (or ideally a small experiment) on whether RAJ, GH, and ES can be integrated with other base quantization frameworks beyond DuQuant. Even a brief analysis of what assumptions tie the current implementation to DuQuant would help readers assess the broader applicability of the approach.
>
> We chose DuQuant as the main base quantizer because it is a strong dense-model PTQ method and has also been adopted as the base quantizer in prior MoE quantization work such as EAQuant. To directly evaluate portability, we conducted an additional experiment by integrating our proposed components into SmoothQuant on OLMoE under W4A4. As shown below, adding ES, RAJ, and GH improves the SmoothQuant baseline from 55.07 to 56.79 average zero-shot accuracy, yielding a +1.72 absolute improvement.
>
> This experiment suggests that the proposed components are not tied to DuQuant. The portability comes from the fact that RAJ and GH only require the full-precision and quantized router logits, while ES relies on per-expert activation and weight statistics, all of which are generally available during PTQ calibration. Therefore, these components can be integrated into other calibration-based quantization frameworks that expose quantized router outputs and allow quantization parameters or smoothing factors to be adjusted. We have added this SmoothQuant experiment and the corresponding portability discussion in Appendix F under **Portability to SmoothQuant**.
>
> | Model | arc_c | arc_e | boolq | hellaswag | openbookqa | rte | winogrande | Average |
> | --- | ---: | ---: | ---: | ---: | ---: | ---: | ---: | ---: |
> | SmoothQuant | 39.68 | 69.99 | 62.29 | 54.00 | 30.20 | 63.90 | 65.43 | 55.07 |
> | SmoothQuant + ES + RAJ + GH | 42.15 | 73.65 | 65.62 | 55.69 | 29.40 | 64.21 | 66.80 | 56.79 |
>
> > **Weakness 2**: Discussion of when gains are smaller. The improvements vary across models. It would be helpful if the authors could provide some intuition for this variation.
>
> We agree that the gain variation across models deserves more discussion. Our interpretation is that the improvement mainly depends on the amount of correctable router--expert mismatch introduced by quantization. When quantization substantially changes the top-$k$ expert set/order or reduces the router logit margin, RAJ and GH have more room to improve routing stability. In contrast, when the baseline already preserves routing decisions reasonably well, or when the remaining degradation is dominated by expert-side weight/activation quantization error, the additional downstream gain becomes smaller.
>
> This trend is also reflected by the Match Score results in Table 4. Under W4A4, RouteQuant improves the Match Score over DuQuant by +3.24 on OLMoE, +3.13 on DeepSeek-MoE, and +2.29 on Qwen3-MoE. The smaller Match Score improvement on Qwen3 is consistent with its smaller downstream accuracy gain. This suggests that RouteQuant remains effective in improving routing consistency across models, but the downstream benefit depends on how much of the quantization-induced degradation is attributable to correctable routing mismatch.
>
> We also note that the reported downstream improvements are relative gains over DuQuant. Therefore, their magnitude can be affected by the strength of the DuQuant baseline on each model. Nevertheless, the Match Score trends provide a model-level explanation beyond this normalization effect. We have incorporated this discussion into Section 4.2 under **Router Consistency (Match Score)**, where we relate the Match Score trends in Table 4 to the downstream accuracy results.

---

> > ### Author Response · Authors · 2026-05-25
> > **Response to Reviewer vudq (2/2)**
> >
> > > **Weakness 3**: Calibration cost. Inference runtime is well-documented (Table 7), but reporting the wall-clock calibration time for RouteQuant compared to baselines would be a useful addition for practitioners.
> >
> > We measured the wall-clock calibration time on OLMoE using the same setup as our main experiments: a single NVIDIA A100 80GB GPU, 256 C4 calibration sequences, and sequence length 2048. We repeated each method five times and report the mean and standard deviation. As shown below, RouteQuant takes 37.34 minutes on average. While this is higher than the DuQuant backbone due to the additional router-related calibration objectives, its calibration time is comparable to strong MoE PTQ baselines such as EAQuant and QESC. Moreover, RAJ and GH are used only during calibration and introduce no inference-time overhead, while ES is computed once from calibration statistics. We have added this comparison in Appendix F under **Calibration Time Comparison** to make the cost--performance trade-off clearer for practitioners.
> >
> > | Method | Calibration Time (mins) |
> > |---|---:|
> > | DuQuant | 15.75 ± 0.32 |
> > | MoEQuant | 22.65 ± 0.46 |
> > | EAQuant | 37.71 ± 0.44 |
> > | QESC | 37.58 ± 0.68 |
> > | RouteQuant | 37.34 ± 0.52 |
> >
> >
> > ---
> >
> > Thank you once again for your valuable feedback. We welcome further questions or suggestions for refinement.

---

### Review · Reviewer_c72F · 2026-05-13

**Summary Of Contributions:**

Summary of Contributions:

- The paper studies post-training quantization for Mixture-of-Experts LLMs, focusing on the interaction between router decisions and quantized expert outputs.

- The key observation is that keeping the router in full precision may be insufficient, since quantized experts can shift activations and affect downstream routing decisions.

- The authors propose RouteQuant, which combines Rank-Aware Jaccard Loss for aligning top-k expert selections, Gap Hinge Loss for preserving router logit margins, and Expert-Aware Smoothing for expert-specific quantization.

- The method is evaluated on OLMoE, DeepSeek-MoE, and Qwen3-MoE under W4A4 and W4A8 settings. The results show consistent improvements in perplexity and average zero-shot accuracy over several PTQ baselines.

Strengths:

- The paper addresses a practical and relevant problem in MoE quantization.
- The motivation around router-expert coupling is reasonable and specific to MoE models.
- The proposed components are technically simple and easy to integrate into a PTQ pipeline.
- The experiments cover multiple MoE backbones, quantization settings, and evaluation tasks.
- The ablation and router consistency analyses provide useful support for the proposed design.

Weaknesses:

- The method is more incremental than fundamental, as the main contribution comes from combining several targeted calibration objectives.
- The connection among RAJ, GH, and ES could be presented more clearly as a unified framework.
- Some performance gains over the strongest MoE-specific baselines are modest, even though the average trend is consistently positive.
- The paper could better clarify when router quantization is expected to help or hurt.

**Audience:**

Yes

**Audience Explanation:**

Yes. The paper should be of interest to part of the TMLR audience, especially readers working on model compression, quantization, efficient LLM inference, and MoE architectures.

The topic is practical and timely, since MoE models are increasingly used but remain difficult to deploy efficiently. The paper also studies a MoE-specific issue, namely the interaction between router behavior and expert quantization, rather than only applying standard dense-model PTQ techniques.

The contribution is more engineering-oriented than conceptually groundbreaking, but the findings are still useful for researchers and practitioners interested in low-bit MoE deployment.

**Claims And Evidence:**

Yes

**Claims Explanation:**

The main claims are mostly supported. The paper evaluates RouteQuant on three MoE models under W4A4 and W4A8, and shows consistent improvements in average accuracy and perplexity over the main PTQ baselines.

The router consistency analysis, perfect-match experiment, and ablation study further support the motivation that routing mismatch matters and that ES, RAJ, and GH each contribute.

That said, some gains over the strongest MoE-specific baselines are modest, so the claims should be stated with some nuance.

**Requested Changes:**

Critical for acceptance:

- Clarify the main claim about router quantization. The paper argues that keeping the router in full precision is suboptimal, but the main setting uses W8A8 routers unless otherwise specified. The authors should more clearly distinguish between router quantization, router-aware calibration, and aggressive low-bit router quantization.

- Clarify the theoretical claim around gap preservation and top-k stability. The current presentation should make explicit what assumptions are needed for gap preservation to imply stable expert ordering or reduced rank flipping.

Would strengthen the work:

- Improve the transition into Section 3.1. The method section introduces many symbols immediately, but the motivation for decomposing the selected expert sets could be explained more intuitively.

- Better explain why RAJ, GH, and ES form a unified framework rather than three separate calibration heuristics.

- Add a more detailed discussion of when router quantization helps or hurts, since the paper itself notes that the improvement is not universal.

- Report more stability evidence, such as calibration seed variance or confidence intervals, especially because some gains over the strongest MoE-specific baselines are modest.

---

> ### Author Response · Authors · 2026-05-25
> **Response to Reviewer c72F (1/3)**
>
> Thank you for the thoughtful and constructive review. We appreciate your positive assessment of the practical relevance of the problem, the MoE-specific router–expert coupling motivation, and the experimental, ablation, and router-consistency analyses. Your comments on clarifying the router-quantization claim, the assumptions behind gap preservation, and the unified role of RAJ, GH, and ES are very helpful. We address these points below and have revised the manuscript to present the claims more precisely and with appropriate nuance.
>
> ---
>
> > **Weakness 1**: Clarify the main claim about router quantization. The paper argues that keeping the router in full precision is suboptimal, but the main setting uses W8A8 routers unless otherwise specified. The authors should more clearly distinguish between router quantization, router-aware calibration, and aggressive low-bit router quantization.
>
> We agree that the main claim should more clearly distinguish vanilla router quantization, router-aware calibration, and aggressive low-bit router quantization. Our intended claim is not that simply quantizing the router is always better than keeping it in full precision. Rather, our claim is that when the experts and MoE blocks are quantized, the router should not be treated as an isolated frozen full-precision module; instead, it should be included in a router-aware calibration procedure to preserve stable routing under quantization.
>
> First, for plain router quantization, where no additional router-aware objective is introduced, we provide the comparison in Figure 2. The results show that simply quantizing the router does not always outperform keeping the router in full precision. This supports our main motivation: router quantization itself is not automatically beneficial, and a more careful router-aware quantization strategy is needed.
>
> Second, our main setting focuses on router-aware calibration with W8A8 routers. In this setting, the router is quantized, but the key contribution is not merely applying W8A8 quantization to the router. Rather, the contribution is to calibrate the router together with the quantized MoE components so that the routing decision better matches the behavior of the quantized experts. We report this setting in Table 3.
>
> Third, we further study aggressive low-bit router quantization, where the router activation is reduced to W4A8. This setting is more challenging because routing decisions are sensitive to small perturbations in router logits and top-$k$ expert selection. We report these results in Table 6 to evaluate whether our method remains effective under a stricter router-quantization regime.
>
> We also agree that Tables 3 and 6 do not isolate the router effect as directly as they could. To address this, we provide an additional summarized comparison below that focuses specifically on the impact of router quantization under controlled settings. The results show that our approach further improves upon vanilla router quantization. We have added this clarification and the summarized analysis in Appendix F under **Controlled Comparison of Router Precision**.
>
>
> | Model | arc_c | arc_e | boolq | hellaswag | openbookqa |   rte | winogrande |     Average |
> | -------------------- | ----: | ----: | ----: | --------: | ---------: | ----: | ---------: | ----------: |
> | router(W4A8)  |  40.70 | 74.12 | 66.15 |     55.99 |       32.40 | 67.87 |      66.85 | 57.73 |
> | router(W8A8)  | 43.17 | 72.85 | 65.57 |     56.21 |         32.00 | 69.31 |      65.67 | 57.83 |
> | router(FP16) | 40.27 | 72.43 | 65.38 |     55.96 |       31.60 | 70.42 |      65.82 | 57.41 |
>
> > **Weakness 2**: Clarify the theoretical claim around gap preservation and top-k stability. The current presentation should make explicit what assumptions are needed for gap preservation to imply stable expert ordering or reduced rank flipping.
>
> We agree that the connection between gap preservation and top-$k$ stability should be stated more carefully. In the revision, we clarify that Proposition 2 is itself a deterministic sufficient condition for preserving expert ordering. Specifically, for two experts $i$ and $j$, if the full-precision router satisfies $r_i(x) > r_j(x)$, and the quantized router preserves a non-shrunk gap in the same direction, i.e.,
> $$
> \hat r_i(x) - \hat r_j(x) \ge r_i(x) - r_j(x) > 0,
> $$
> then the ordering between these two experts is preserved, since $\hat r_i(x) > \hat r_j(x)$. Therefore, no additional assumption is required for the proposition itself; it directly states a sufficient condition under which rank flipping cannot occur for that pair.
>
> We also clarify that the proposed GH loss is not a guarantee that Proposition 2 holds for every token and every expert pair. Rather, GH is a practical objective designed to encourage the condition in Proposition 2 by penalizing margin shrinkage between important router logits, especially around the top-$k$ decision boundary. We have revised the beginning of Section 3.3 to make this clear.

---

> > ### Author Response · Authors · 2026-05-25
> > **Response to Reviewer c72F (2/3)**
> >
> > > **Weakness 3**: Improve the transition into Section 3.1. The method section introduces many symbols immediately, but the motivation for decomposing the selected expert sets could be explained more intuitively.
> >
> > We agree that the previous transition introduced notation too abruptly. We have revised the beginning of Section 3.1 to first explain the intuition behind decomposing the selected expert sets before presenting the formal notation.
> >
> > Specifically, the revised text now explains that after quantization, the quantized router may not select exactly the same top-$k$ experts as the FP16 router. The two selected sets can therefore be decomposed into: (i) shared experts that remain selected in both models, (ii) dropped experts that were selected by the FP16 router but removed after quantization, and (iii) newly selected experts that enter the quantized top-$k$ set. This decomposition separates two sources of routing error: changes in the outputs of shared experts and changes caused by replacing dropped experts with newly selected ones.
> >
> > We believe this transition better motivates the notation and makes the subsequent error decomposition easier to follow.
> >
> > > **Weakness 4**: Better explain why RAJ, GH, and ES form a unified framework rather than three separate calibration heuristics.
> >
> > We agree that the method overview should better explain why RAJ, GH, and ES form a unified framework rather than three independent calibration heuristics.
> >
> > In our method, RAJ and GH address the routing-mismatch terms from complementary perspectives. RAJ directly reduces top-$k$ expert-set mismatch by aligning the selected expert identities between the FP16 and quantized routers. GH further stabilizes this selection by preserving router-logit margins, especially near the top-$k$ boundary, thereby reducing rank flipping under quantization noise. ES addresses the expert-output distortion term by applying expert-specific smoothing to reduce per-expert activation/weight quantization error.
> >
> > Therefore, the three components correspond to different terms in the same router--expert error decomposition: RAJ controls whether the correct experts are selected, GH controls whether their selection and ordering remain stable, and ES improves the quantized outputs produced by those selected experts. We have revised the Section 1 to make this connection explicit.
> >
> > > **Weakness 5**: Add a more detailed discussion of when router quantization helps or hurts, since the paper itself notes that the improvement is not universal.
> >
> > We agree that the paper should more clearly discuss when router quantization helps or hurts. Router quantization is not universally beneficial by itself. It can hurt when the router logits have small top-$k$ boundary margins, since small quantization perturbations may flip the selected experts or their order. It can also bring limited gains when the baseline already preserves routing decisions well, or when the dominant error comes from expert-side weight/activation quantization rather than routing mismatch. In addition, overly aggressive router quantization may introduce extra noise, especially for models with a larger routing space or more experts, where the top-$k$ decision can be more sensitive. We have added this discussion in Section 1.

---

> > > ### Author Response · Authors · 2026-05-25
> > > **Response to Reviewer c72F (3/3)**
> > >
> > > > **Weakness 6**: Report more stability evidence, such as calibration seed variance or confidence intervals, especially because some gains over the strongest MoE-specific baselines are modest.
> > >
> > > Thank you for the suggestion. We agree that stability evidence is important, especially when some gains over strong MoE-specific baselines are modest. Specifically, we evaluated OLMoE using C4 calibration data with 256 samples and a sequence length of 2048 tokens, and repeated the calibration process with five different random seeds.
> > >
> > > We report the mean and standard deviation across these five runs. The results are very close to the main experimental results, indicating that the observed improvements are stable and not caused by a particular calibration sample or random seed. We have included this analysis in Appendix F under **Robustness Across Calibration Seeds** to better support the robustness of RouteQuant.
> > >
> > > | Method     |        arc_c |        arc_e |        boolq |    hellaswag |   openbookqa |          rte |   winogrande |            Avg |
> > > | ---------- | :----------- | :----------- | :----------- | :----------- | :----------- | :----------- | :----------- | :------------- |
> > > | FP16       |        46.59 |        77.10 |        70.09 |        58.47 |        32.60 |        71.12 |        68.51 |          60.64 |
> > > | DuQuant    | 40.87 ± 1.36 | 72.33 ± 0.57 | 63.98 ± 1.04 | 54.87 ± 0.62 | 30.20 ± 0.66 | 60.47 ± 0.76 | 64.55 ± 0.49 |      55.32 (-) |
> > > | MoEQuant   | 40.66 ± 0.68 | 73.58 ± 1.41 | 66.29 ± 0.55 | 55.77 ± 0.89 | 29.80 ± 0.43 | 62.21 ± 1.25 | 63.56 ± 0.72 | 55.98 (+1.19%) |
> > > | EAQuant    | 41.45 ± 1.09 | 73.52 ± 0.46 | 66.86 ± 1.32 | 56.11 ± 0.61 | 30.60 ± 0.28 | 62.28 ± 0.28 | 65.38 ± 1.16 | 56.60 (+2.31%) |
> > > | QESC       | 41.32 ± 0.44 | 73.38 ± 0.74 | 65.59 ± 1.21 | 55.87 ± 0.44 | 30.00 ± 0.76 | 65.72 ± 1.03 | 65.88 ± 0.52 | 56.82 (+2.71%) |
> > > | RouteQuant | 42.86 ± 0.75 | 73.24 ± 0.58 | 66.34 ± 0.72 | 56.31 ± 0.16 | 32.40 ± 0.78 | 68.29 ± 1.66 | 65.48 ± 0.84 | 57.85 (+4.57%) |
> > >
> > > ---
> > >
> > > Thank you once again for your valuable feedback. We welcome further questions or suggestions for refinement.

---

### Review · Reviewer_c6QN · 2026-05-19

**Summary Of Contributions:**

This paper studies post-training quantization for Mixture-of-Experts language models. The main focus is the router. The authors argue that keeping the router in full precision is not always enough, because quantized experts can change the hidden states and shift the behavior seen by later routers. Based on this idea, the paper proposes RouteQuant, a calibration-only PTQ method for MoE models.

RouteQuant has three main parts. First, it uses an Expert-Aware Smoothing Factor to give each expert its own smoothing scale. Second, it introduces Rank-Aware Jaccard Loss to align the top-k expert choices between the full-precision and quantized models. Third, it uses Gap Hinge Loss to preserve the score gaps between nearby experts, so that rank flips are less likely.

The paper has several strengths. The problem is important, since MoE models are widely used and low-bit inference is useful for deployment. The paper gives a clear view of router rank instability and proposes losses that directly target top-k expert choices and score margins. The experiments are also broad: the authors evaluate OLMoE, DeepSeek-MoE, Qwen3-MoE, and a multimodal MoE model, under several bit settings. The reported results show consistent gains over the selected baselines.

The main weakness is that the strongest claim of the paper is not yet fully supported. The paper claims that freezing the router is suboptimal because expert quantization shifts expert outputs and a quantized router can better co-adapt with quantized experts. However, the current evidence mostly shows that better router consistency helps downstream accuracy. It does not directly show that a full-precision router is systematically worse under quantized experts. Also, the proposed router losses still align the quantized router to the full-precision router, which seems closer to preserving FP16 routing than learning a new routing policy adapted to quantized experts. The paper is promising, but the main narrative needs stronger support and clearer positioning.

**Additional Comments:**

I like the direction of this paper. The problem is important, and the proposed losses are intuitive and useful. The experiments also show encouraging results. My main concern is not the usefulness of the method, but the strength of the main claim.

**Audience:**

Yes

**Audience Explanation:**

This paper addresses an important and timely problem. Efficient inference for MoE models is highly relevant to the TMLR audience, especially as MoE architectures become more common in large language models and multimodal models.

**Broader Impact Concerns:**

I do not see major ethical concerns specific to this paper.

**Claims And Evidence:**

No

**Claims Explanation:**

Some empirical claims are well supported. The tables show that RouteQuant improves perplexity and accuracy over the chosen baselines in many settings. The evaluation is broad, and the improvements are generally consistent. These results are useful and suggest that the method is effective.

However, the strongest claims are not yet supported by clear enough evidence. The paper claims that freezing the router is suboptimal because expert quantization changes expert outputs, and the router should be quantized to co-adapt with the quantized experts. This is an interesting idea, but the paper does not directly prove this chain of reasoning.

- First, the paper does not directly measure the claimed expert-output shift. It would be useful to see layer-wise or token-wise comparisons between full-precision expert outputs and quantized expert outputs. For example, the authors could report L2 error, cosine similarity, MoE block output error, or how expert output changes affect the next router logits. The current figures mainly show routing rank changes, not the expert-output shift itself.
- Second, the evidence that a quantized router is better than a frozen router is not fully controlled. In Table 8, the “w/o router” setting seems to use ES only with a full-precision router, while the “w/ router” setting uses the full RouteQuant method with ES, RAJ, GH, and a quantized router. Therefore, the improvement may come from the extra RAJ/GH objectives, not from router quantization itself. A stronger comparison would keep all other factors fixed and vary only the router precision.
- Third, the method and the story are not fully aligned. The paper says that the router should co-adapt to quantized experts, but RAJ and GH mainly force the quantized router to match the full-precision router’s top-k set and margins. This seems to preserve FP16 routing rather than adapt routing to quantized experts. If FP16 routing is still the target, the paper should explain why freezing the FP16 router is not enough. If quantized-expert-aware routing is the target, the paper should show that RouteQuant learns a better routing behavior than FP16 routing under quantized experts.
- Fourth, the paper should be more careful in describing prior work. The paper suggests that prior methods commonly keep the router in full precision. However, in practice, some MoE quantization methods or toolkits quantize the router, while others keep it in full precision. The paper should avoid presenting “freezing the router” as the only common setting. A more balanced framing would make the contribution clearer.

**Requested Changes:**

- Provide direct evidence for the expert-output shift hypothesis: The paper’s main motivation is that expert quantization shifts expert outputs and makes a frozen router suboptimal. This claim needs direct evidence. The authors should measure how much expert outputs change after quantization, using metrics such as L2 distance, cosine similarity, MoE block output error, or next-layer router logit shift.
- Add a controlled experiment that isolates router quantization: Table 8 does not isolate the effect of router quantization. The authors should compare FP16 router, W8A8 router, and W4A8 router under the same expert quantization setting, the same calibration data, and the same objectives. This would show whether the gain comes from router quantization itself or from the added RAJ/GH objectives.
- Clarify whether the goal is to preserve FP16 routing or adapt to quantized experts: The current method aligns the quantized router to the FP16 router, but the paper claims that the router should co-adapt with quantized experts. These two ideas are not the same. The authors should explain this clearly. If FP16 routing is the target, then the “freezing is suboptimal” claim needs more care. If adapted routing is the target, the method should include evidence that RouteQuant finds better routing for quantized experts, not just routing closer to FP16.
- Improve the novelty discussion relative to EAQuant, QESC, and other MoE PTQ methods: The paper should give a clearer comparison to prior router-aware or expert-selection-aware quantization methods. A table would help. It should compare whether each method quantizes the router, aligns router logits, aligns top-k sets, models rank order, models logit gaps, uses expert-specific smoothing, and supports multimodal MoEs.
- Avoid overstating that prior methods generally freeze routers: The paper should use a more careful statement. Some methods and systems keep routers in full precision, while others quantize routers. The contribution should be framed around improving router-quantized MoE PTQ, not around replacing a single standard practice.

---

> ### Author Response · Authors · 2026-05-25
> **Response to Reviewer c6QN (1/2)**
>
> Thank you for your thoughtful and constructive review. We appreciate your recognition of the importance of efficient MoE inference and the usefulness of RouteQuant’s proposed losses and empirical results. We also agree that the original manuscript should better support and clarify the main motivation. In the revision, we have added direct evidence for quantization-induced expert-output shifts, provided controlled router-precision ablations under fixed settings, and clarified the relationship between preserving FP16 routing consistency and adapting the router under quantized experts. We also refined the novelty discussion and use more careful wording when comparing with prior MoE PTQ methods.
>
> > **Weakness 1**: Provide direct evidence for the expert-output shift hypothesis: The paper’s main motivation is that expert quantization shifts expert outputs and makes a frozen router suboptimal. This claim needs direct evidence. The authors should measure how much expert outputs change after quantization, using metrics such as L2 distance, cosine similarity, MoE block output error, or next-layer router logit shift.
>
> We agree that Figure 1 mainly provides routing-level evidence, while a direct measurement of expert-output shift would strengthen the motivation. We therefore added a new analysis in Appendix E. Specifically, on OLMoE under W4A4, we measure the L2 distance between FP16 expert outputs and their quantized counterparts for each decoder layer, using the same input tokens and averaging over tokens and selected experts.
>
> The results show a clear quantization-induced expert-output shift. The L2 distance is relatively small in early layers but increases steadily with depth, becoming much larger in later MoE layers. This suggests that expert quantization introduces perturbations that accumulate across layers rather than remaining local. We further measure the next-layer router logit shift and observe the same increasing trend, indicating that shifted expert outputs propagate to downstream router inputs and can make a frozen router mismatched with the quantized expert pathway.
>
> We have included these results and figures in Appendix E of the revised manuscript. Since the rebuttal format is text-only, we summarize the key finding here: under the OLMoE W4A4 setting, both expert-output L2 distance and next-layer router-logit L2 distance increase after quantization, directly supporting the expert-output shift hypothesis.
>
>
> > **Weakness 2**: Add a controlled experiment that isolates router quantization: Table 8 does not isolate the effect of router quantization. The authors should compare FP16 router, W8A8 router, and W4A8 router under the same expert quantization setting, the same calibration data, and the same objectives. This would show whether the gain comes from router quantization itself or from the added RAJ/GH objectives.
>
> We agree that the original Table 8 did not fully isolate the effect of router quantization. We have revised Table 8 to compare FP16, W4A8, and W8A8 routers under the same expert quantization setting, calibration data, and ES configuration. Specifically, we compare: (i) an FP16 router with ES only; (ii) W4A8/W8A8 routers with ES only, which isolates the effect of router quantization itself; and (iii) W4A8/W8A8 routers with ES+RAJ/GH, which measures the contribution of our router-alignment objectives. The revised Table 8 shows that plain router quantization alone brings mixed or marginal changes: it slightly degrades OLMoE and DeepSeek, and has little effect on Qwen3. In contrast, adding RAJ/GH consistently improves performance over both the FP16-router baseline and the vanilla quantized-router setting across all models and both router precisions. This confirms that the gains mainly come from the proposed router-alignment objectives, rather than router quantization itself.
>
>
> | Model        | Router    | ES | RAJ/GH | Avg. ↑      |
> | ------------ | --------- | -- | ------ | :---------- |
> | **OLMoE**    | FP16      | ✓  | ✗      | 57.41       |
> |              | W4A8/W8A8 | ✓  | ✗      | 56.88/56.96 |
> |              | W4A8/W8A8 | ✓  | ✓      | 57.73/57.83 |
> | **DeepSeek** | FP16      | ✓  | ✗      | 55.23       |
> |              | W4A8/W8A8 | ✓  | ✗      | 54.84/54.95 |
> |              | W4A8/W8A8 | ✓  | ✓      | 55.50/55.59 |
> | **Qwen3**    | FP16      | ✓  | ✗      | 63.52       |
> |              | W4A8/W8A8 | ✓  | ✗      | 63.51/63.58 |
> |              | W4A8/W8A8 | ✓  | ✓      | 63.87/63.95 |

---

> > ### Author Response · Authors · 2026-05-25
> > **Response to Reviewer c6QN (2/2)**
> >
> > > **Weakness 3**: Clarify whether the goal is to preserve FP16 routing or adapt to quantized experts: The current method aligns the quantized router to the FP16 router, but the paper claims that the router should co-adapt with quantized experts. These two ideas are not the same. The authors should explain this clearly. If FP16 routing is the target, then the “freezing is suboptimal” claim needs more care. If adapted routing is the target, the method should include evidence that RouteQuant finds better routing for quantized experts, not just routing closer to FP16.
> >
> > We agree that the distinction should be made clearer. RouteQuant does not aim to discover an entirely new routing policy that is task-optimal for the quantized experts. Instead, our goal is to preserve the FP16 routing behavior as much as possible under the perturbed computation induced by quantization.
> >
> > The reason a frozen FP16 router can still be suboptimal is that the router weights may remain full precision, but the router inputs are no longer identical to those in the FP16 model. Quantization of previous layers and MoE blocks can perturb the hidden states passed to later routers, causing even a frozen FP16 router to produce routing decisions that deviate from the original FP16 routing behavior. Therefore, simply keeping the router weights in full precision does not necessarily preserve FP16 routing.
> >
> > Under this view, "co-adaptation" means calibrating the router under the quantized activation distribution so that it remains compatible with the quantized model while recovering the FP16 routing behavior as closely as possible. RAJ aligns the top-$k$ expert identities, and GH preserves the relevant router-logit margins, both using the FP16 router outputs as the reference. Thus, our target is FP16 routing consistency under quantized inputs, rather than unconstrained adaptation to a new routing policy.
> >
> > We have revised Section 1 (Introduction) to better explain this.
> >
> >
> > > **Weakness 4**: Improve the novelty discussion relative to EAQuant, QESC, and other MoE PTQ methods: The paper should give a clearer comparison to prior router-aware or expert-selection-aware quantization methods. A table would help. It should compare whether each method quantizes the router, aligns router logits, aligns top-k sets, models rank order, models logit gaps, uses expert-specific smoothing, and supports multimodal MoEs.
> >
> > We agree that the novelty discussion should more clearly position RouteQuant relative to prior MoE PTQ methods. We added a comparison table covering MoEQuant, QESC, EAQuant, and RouteQuant, focusing on the aspects suggested by the reviewer.
> >
> > | Method | Q-Router | Logit Align. | Top-$k$ Align. | Rank | Gap | ES | MM Eval. |
> > | --- | :-: | :-: | :-: | :-: | :-: | :-: | :-: |
> > | MoEQuant | ✗ | ✗ | ✗ | ✗ | ✗ | ✗ | ✗ |
> > | QESC | ✗ | ✓ | △ | ✗ | ✗ | ✗ | ✗ |
> > | EAQuant | ✓ | ✓ | △ | ✗ | ✗ | △ | ✗ |
> > | RouteQuant | ✓ | ✓ | ✓ | ✓ | ✓ | ✓ | ✓ |
> >
> > Here, Q-Router denotes whether the router is quantized; Logit Align. denotes router-logit or distribution alignment; Top-$k$ Align. denotes explicit top-$k$ expert-set alignment; Rank and Gap denote whether rank order and inter-expert logit gaps are explicitly modeled; ES denotes expert-specific smoothing; and MM Eval. denotes evaluation on multimodal MoEs. The symbol △ indicates that the aspect is only partially or indirectly addressed.
> >
> > Compared with prior MoE PTQ methods, RouteQuant explicitly models the two routing factors identified in our analysis: top-$k$ expert identity/order through RAJ and inter-expert logit-gap preservation through GH. In addition, RouteQuant introduces expert-specific smoothing and evaluates the framework on multimodal MoEs, further distinguishing it from existing MoE PTQ methods. We have added this analysis as a new Appendix D section, **Comparison with Existing MoE PTQ Methods**, in the revised manuscript.
> >
> >
> > > **Weakness 5**: Avoid overstating that prior methods generally freeze routers: The paper should use a more careful statement. Some methods and systems keep routers in full precision, while others quantize routers. The contribution should be framed around improving router-quantized MoE PTQ, not around replacing a single standard practice.
> >
> > We agree that the original wording may overstate the extent to which prior methods freeze routers. In the revision, we use a more careful framing: some PTQ systems keep routers in full precision due to their small parameter cost and sensitivity, while recent MoE PTQ methods may also quantize or calibrate routers. Therefore, our contribution should not be interpreted as replacing a universal router-freezing practice. Instead, we have revised the framing in Section 2.2 to present RouteQuant as improving router-quantized MoE PTQ.
> >
> > ---
> >
> > Thank you once again for your valuable feedback. We welcome further questions or suggestions for refinement.

---

### Decision · Action_Editor_3H1Z · 2026-07-02

**Recommendation:** Accept as is

**Audience:**

Yes

**Audience Explanation:**

Explain your answer above
Yes. The paper addresses post-training quantization for Mixture-of-Experts language models, which is a timely and practically important topic for researchers working on model compression, efficient inference, quantization, and MoE architectures. The work studies a MoE-specific issue, the interaction between expert quantization, hidden-state perturbations, and downstream router decisions, rather than simply applying dense-model PTQ methods unchanged. Although the contribution is not conceptually groundbreaking, the findings and calibration objectives are likely to be useful to part of the TMLR audience interested in low-bit deployment of MoE models.

**Claims And Evidence:**

Yes

**Claims Explanation:**

The revised submission provides sufficient evidence for its main claims. The paper evaluates RouteQuant across multiple MoE backbones and quantization settings, including OLMoE, DeepSeek-MoE, and Qwen3-MoE under W4A4 and W4A8, and reports consistent improvements in perplexity and zero-shot accuracy over the considered PTQ baselines. The ablations, router-consistency analyses, and theoretical decomposition support the proposed roles of Expert-Aware Smoothing, Rank-Aware Jaccard Loss, and Gap Hinge Loss.

The main concern raised during review was that the original manuscript overstated the “beyond freezing the router” narrative and did not fully isolate whether the gains came from router quantization itself or from the proposed router-aware calibration objectives. In the revision, the authors added direct evidence of quantization-induced expert-output and next-layer router-logit shifts, provided controlled comparisons of FP16, W4A8, and W8A8 router settings under fixed calibration conditions, and clarified that the contribution is router-aware calibration to preserve stable routing under quantized activations rather than vanilla router quantization alone. The reviewers judged these revisions sufficient. The remaining limitations are that the gains over strong MoE-specific baselines are moderate and the contribution is primarily incremental/engineering-oriented, but these limitations do not undermine the paper’s soundness.